https://doi.org/10.1038/s42003-021-02485-4　　**OPEN**
# Long range synchronization within the enteric nervous system underlies propulsion along the large intestine in mice

Nick J. Spencer [1✉], Lee Travis[1], Lukasz Wiklendt[2], Marcello Costa[1], Timothy J. Hibberd [1], Simon J. Brookes[1], Phil Dinning[2], Hongzhen Hu[3], David A. Wattchow[4] & Julian Sorensen[1]

How the Enteric Nervous System (ENS) coordinates propulsion of content along the gastrointestinal (GI)-tract has been a major unresolved issue. We reveal a mechanism that explains how ENS activity underlies propulsion of content along the colon. We used a recently developed high-resolution video imaging approach with concurrent electrophysiological recordings from smooth muscle, during fluid propulsion. Recordings showed pulsatile firing of excitatory and inhibitory neuromuscular inputs not only in proximal colon, but also distal colon, long before the propagating contraction invades the distal region. During propulsion, wavelet analysis revealed increased coherence at ~2 Hz over large distances between the proximal and distal regions. Therefore, during propulsion, synchronous firing of descending inhibitory nerve pathways over long ranges aborally acts to suppress smooth muscle from contracting, counteracting the excitatory nerve pathways over this same region of colon. This delays muscle contraction downstream, ahead of the advancing contraction. The mechanism identified is more complex than expected and vastly different from fluid propulsion along other hollow smooth muscle organs; like lymphatic vessels, portal vein, or ureters, that evolved without intrinsic neurons.

[1] Visceral Neurophysiology Laboratory, College of Medicine and Public Health, Centre for Neuroscience, Flinders University, Bedford Park, SA, Australia. [2] Discipline of Gastroenterology, College of Medicine and Public Health, Flinders Medical Centre, Bedford Park, SA, Australia. [3] Department of Anesthesiology, The Center for the Study of Itch, Washington University, St Louis, MO, USA. [4] Discipline of Surgery, College of Medicine and Public Health, Flinders Medical Centre, Bedford Park, SA, Australia. ✉email: nicholas.spencer@flinders.edu.au

In vertebrates, many hollow organs consist of layers of smooth muscle cells, which are rhythmically excitable and can generate propagating contractions that propel fluid over some distance. These include lymphatic vessels[1], ureters[2], urethra[3], some blood vessels[4] including the portal vein[5] and parts of the gastrointestinal (GI) tract[6]. The generation of smooth muscle contraction in these organs occurs via mechanisms either intrinsic to muscle cells themselves, or by distinct pacemaker cells, which induce electrical rhythmicity in smooth muscle[3,7–9]. Indeed, in parts of the GI tract (e.g, stomach and small bowel), distinct pacemaker cells have been identified[8–10]. The electrical rhythmicity that they generate in smooth muscle is largely responsible for propulsion of liquid over short distances[6,11].

**Fig. 1 Simultaneous video imaging and electrophysiological recordings from smooth muscle in isolated mouse colon. a** shows a photomicrograph of the isolated colon and location of two independent extracellular recording electrodes, separated by 1 mm in the mid colon. **b** spatio-temporal map with superimposed simultaneous electrical recordings (red:proximal and blue:distal) both taken from the mid colon. The white band propagating from the proximal to distal colon indicates the propulsive contraction that underlies the propulsion of fluid aborally (see Supplementary Movie 1). **c** shows the raw electrical recordings superimposed. **d** Wavelet coherence showing a peak frequency at ~2 Hz (see yellow band). **e** shows the relative time difference between the two independent electrical recordings. There is a very small temporal difference between EJPs occurring at both recording sites. fi shows the two electrical recordings superimposed on expanded scale, taken from the period represented by the dotted line in **e**. fii shows the recordings from fi on further expanded scale where the temporal coordination of EJPs at both recording sites is apparent.

The GI tract is unique, compared to all other hollow smooth muscle organs, because it evolved with its own independent nervous system, the Enteric Nervous System (ENS)[12–20]. The ENS consists of a complex network of two distinct ganglionated nerve plexuses including both excitatory and inhibitory motor neurons to smooth muscle cells, multiple classes of ascending and descending interneurons[21], and a unique population of intrinsic sensory neurons[13,17,20,22,23]. Animal models of Hirschsprung's disease in humans, that lack large segments of ENS, either die at birth, or soon after birth, due to improper intestinal transit[20,24–28], reflecting the functional importance of the ENS.

During last half of the 20th century, there have been major advances in our understanding of the different classes of neurons in the ENS of vertebrates, particularly with regards to their neurochemical coding, electrical properties, synaptic inputs and projections[12–15,19,20,29–33]. Perhaps the major unresolved issue that has prevailed is how all these different classes of neurons in the ENS are temporally and spatially activated during patterns of motor activity that underlie propulsion along the gut.

Recently, neuronal imaging of the ENS during cyclical neurogenic motor activity in the mouse colon identified that large populations of enteric neurons fired in coordinated and repetitive bursts, which generate excitatory junction potentials (EJPs) at the same rate in the smooth muscle[27]. Thus, the discharge pattern of EJPs in the smooth muscle can provide us with a direct insight into the ENS firing rate[27]. Such motor activity, originally described in the mouse colon as colonic migrating motor complexes (CMMCs)[34–41], is present in other species such as human colon[42,43] and guinea-pig colon[44]. A recent consensus paper suggested that this pattern should be generically called "Colonic Motor Complexes" (CMCs)[45]. We took advantage of a recent technical advance from our laboratory[44], enabling smooth muscle electrical recordings along the length of colon to be correlated with dynamic changes in diameter during propulsion. This allows the patterns of neuronal activity in the ENS that underlie propulsion to be inferred.

The findings reveal that colonic propulsion involves synchronization of large enteric neural assemblies over long distances that includes the suppression of descending excitation. This is unmasked by blockade of nitric oxide synthesis. This study uncovers a level of complexity and sophistication in the ENS beyond what has been previously presumed.

## Results
### Ensemble activation of excitatory and inhibitory neurons in the ENS drives coordination and propulsion
Smooth muscle electrical activity was recorded from two separate sites varying in their longitudinal separation along the colon, during fluid distension-evoked activity. Initially, electrical recordings were made from the smooth muscle in the mid colon, with the two electrodes separated by 1 mm in the longitudinal axis (Fig. 1a). In response to fluid infusion into the proximal colon (Fig. 1a), a contraction was initiated in the proximal colon that propagated aborally (see Supplementary Movie 1; 80 of 87 distensions, $N = 13$). In 8% of trials, the contraction commenced in the mid colon

and then propagated aborally (8% in the mid colon; 7 of 87 distensions, $N = 13$).

Before the distension-evoked contraction was elicited in the proximal colon, a repetitive discharge of EJPs was recorded in the smooth muscle of the mid colon at $1.57 \pm 0.36$ Hz (range: 1.2–1.9 Hz; $N = 13$; measured by Continuous Wavelet Transform (CWT) with a Morlet wavelet; See Fig. 1b, c and Supplementary Movie 1). Some EJPs reached action potential threshold (Fig. 1b, Supplementary Movie 1) especially in the mid colon (see electrode position in Fig. 1a, Supplementary Movie 1). Electrical recordings made 1 mm apart in the mid colon showed that EJPs were always highly synchronized at these closely spaced sites (Fig. 1 and Supplementary Movie 1 and Supplementary Movie 2). This is further exemplified in Supplementary Movie 2, which shows an expanded period of Supplementary Movie 1, with EJPs occurring at the same rates, with similar amplitudes and time courses, at both electrodes. The degree of synchronization between EJPs was quantified at increasing electrode separation distances from 1 to 30 mm (when one electrode was in proximal-mid colon and the other in distal colon) (Figs. 2a, 3, Supplementary Movie 3). The median frequency of EJPs that occurred in response to fluid distension is for the range of distances considered is shown in Fig. 4a. The degree of synchronization of electrical activities between the proximal and distal colon was quantified using Wavelet Coherence (WCOH) and Frequency coordination; see the Methodology Section for analysis details and Supplementary Table 1 for analysis results. A major observation was that for all distances considered, during the onset of the CMC contraction in the proximal colon, EJPs were found to discharge synchronously between the proximal and distal colon long before the contraction wavefront had propagated into the distal colon (see Fig. 1a, Supplementary Movie 3 and Supplementary Movie 4).

### Colonic motor complexes without fluid propulsion and effects of nicardipine, atropine and tetrodotoxin
The results above revealed that during aboral propulsion of fluid, there was an increase in temporal coordination of EJPs in the smooth muscle across at least 30 mm of colon. It might be argued that EJP synchronization along the length of colon was induced by fluid movement within the lumen, driven by a neuromechanical loop phenomenon that has been identified recently[46]. To test this, colonic distension was imposed by inserting a metal rod (2.4 mm diameter) through the lumen, inducing maintained distension (Fig. 5a), but without permitting propulsion of luminal content. Under these conditions, maintained distension-evoked CMCs that were also associated with cyclical bursts of repetitive EJPs that were also found to temporally coordinate between the proximal and distal colon (Fig. 5b, c). The mean discharge frequency of EJPs during bursts was $1.7 \pm 0.4$ Hz (range: 1.3–2.2; $N = 13$; measured by CWT; Figs. 4b, 5c-e). When nicardipine (2 μM) was applied, to paralyse the smooth muscle, the coordinated discharges of EJPs persisted (at $1.8 \pm 0.5$ Hz; range: 1.4–2.4 Hz; $N = 8$; measured by CWT; Figs. 4b, 5c), even temporally coordinated over an electrode separation distance of 50 mm (Figs.6, 7). This confirms that neither tension changes in smooth muscle

## 1mm electrode separation

## 30mm electrode separation

nor propulsion of content were responsible for temporal synchronization of ENS activity underlying EJPs.

Next, we investigated the effects of ENS activity during CMCs, after EJPs were abolished with atropine. Addition of atropine (3 µM) did not abolish CMCs, it simply abolished the EJPs occurring during CMCs, and revealed inhibitory junction potentials (IJPs) (Supplementary Fig.1), which were also

temporally and spatially coordinated when electrodes were separated by 30 mm (Figs. 6, 7). In the presence of atropine, IJPs discharged at a mean rate of $0.9 \pm 0.3$ Hz (Range = 0.6–1.2; $N = 13$; measured by CWT; Fig. 4b), which is lower than the frequency of EJPs without atropine.

A comparative analysis of the degree of temporal coordination between EJPs or IJPs (in atropine) over 1–50 mm electrode

**Fig. 2 Characteristics of electrical activities in smooth muscle over 1 mm and 30 mm electrode separation distances.** Plots **a**–**K** and **l**–**v** pertain to analysis of fluid distension at 1 mm and 30 mm electrode separation respectively, following fluid distension. Plots **a**, **l**: These plots contain the raw data for the proximal and distal colons, represented by the red and blue lines respectively. Plots **b**, **m**: These plots contain the WCOH (absolute value) between the proximal and distal colons. In both the 1 mm and 30 mm examples, the period of increased activity at ~2 Hz in plots **a**, **l** is also reflected in the WCOH. Plots **c**, **n** These plots contain the CWT (absolute value) for the proximal colon, and plots **d**, **o** for the distal colon. The increased activity at ~2 Hz in plots **a**, **l** is reflected in these plots at ~2 Hz. Plot **e**, **p** The red and blue lines are traces of bandpass filtered data (filter limits 0.5 and 3.5 Hz) for the proximal and distal ends respectively. The plots are zoomed in during the period of increased activity at ~2 Hz. The vertical bars denote the peaks of the oscillations; in the 1 mm example, the peaks are almost coincident, whereas in 30 mm example, these peaks will usually be separated further. Plot **f**, **g**, **q**, **r** In both plots, a window (length 3 sec) has been slid over the bandpass filtered data in plots **e**, **p**. In each window instance, the likelihood that the data have a particular frequency is calculated; these plots display the log-likelihood. The frequency range for which the likelihood is calculated is given by the y-axis—in these examples from 1 to 2.8 Hz. Plot **h**, **s** These plots overlay the frequency maximizing the log-likelihood function for the proximal (plots **f**, **q**), and distal colon (plots **q**, **r**). In these plots, the frequencies maximizing the likelihood functions are close during propulsion and diverge elsewhere. Plot **i**, **t** In these plots, a window (length 3 sec) has been slid over the bandpass filtered data in plots **e**, **p**. In each window instance, the cross correlation between the proximal and distal colons is calculated, which these plots display. During propulsion, the cross correlation is periodic with periodicity ~0.5 seconds, corresponding to a frequency of ~2 Hz. Moreover, the periodicity of cross correlation increases with time, which is consistent with the decrease in frequency estimates displayed in plots **h**, **s**. Plots **j**, **k**, **u**, **v** In these plots, a window (length 3 sec) has been slid over the bandpass filtered data in plots **e**, **p**. In each window instance, the auto-correlation function is calculated; plots **j**, **u** contains the auto-correlation function for the proximal colon, and plots **k**, **v** the auto-correlation function for the distal colon. During the period of activity, the auto-correlation functions are both periodic with the same periodicity, which is consistent with the frequency estimates displayed in plots **h**, **s**.

separation was performed using the same measures of coordination as that for fluid distension (WCOH and frequency coordination). We analysed four conditions: (i) when preparations were bathed with normal Krebs solution, (ii) nicardipine and (iii) nicardipine and atropine. The results are presented in Supplementary Table 2 and illustrated in Fig. 6 and 7. The study found that at all electrode separation distances, increased temporal coordination occurred between EJPs (in Krebs and nicardipine) and IJPs (in atropine) across the proximal to distal colonic regions, where recordings were made.

When CMCs and all propulsion was abolished with TTX, myogenic bursts of rhythmic smooth muscle electrical activity were recorded every $31.5 \pm 2.2$ s, with a mean duration of spike bursts of $17.5 \pm 1.6$ s ($N = 6$). In TTX, no ~2 Hz peak band of action potential firing occurred. Rather, activity was observed at ~1 Hz, which is illustrated in the CWT, maximum likelihood and auto-correlation intensity plots in Fig. 8. Applying the same methodology to TTX as for nicardipine, Krebs solution and atropine, it was found that TTX shows a statistically significant increase in WCOH and frequency coordination up to 7 mm. However, after 7 mm, there is no statistically significant increase in WCOH and frequency coordination. More importantly, during activity, at all distances, the WCOH and frequency coordination measures for TTX, was less than for recordings made in nicardipine, Krebs or atropine; the $p$ values for the one-sided t-tests are given in Supplementary Table 3.

The apparent propagation direction of CMCs under maintained colonic distension (with a metal rod) differed to the directionality of fluid propulsion. With the rod in the lumen, 11% of CMCs propagated from distal to proximal colon or occurred simultaneously along the colon (Supplementary Fig. 2; $N = 6$). Removal of the extracellular electrodes from the colon did not change the directionality of propagation, suggesting it was not affected by the mechanical interaction with suction electrodes. Hence, fluid distension elicits a clearly polarized, neurally dependent aboral propulsion, whereas colonic distension maintained by a rod induces more variable propagation of CMCs (Supplementary Fig. 2).

**Control of inhibitory neuromuscular transmission on activation of ENS circuitry.** Of great interest to us was identifying the mechanism responsible for the temporal delay in propagation of smooth muscle contraction during aboral propulsion of fluid. We reasoned that since inhibitory motor neurons in the ENS project

largely in an aboral direction[30,47–49] release of inhibitory neurotransmitter(s) onto smooth muscle downstream could theoretically suppress muscle excitation. Therefore, we sought to determine if there was a preferential change in excitability of the smooth muscle along the length of colon, after blockade of the major inhibitory neurotransmitter, nitric oxide (NO). Addition of L-NOARG enhanced EJPs distal to the contraction, so that they often reached the threshold for action potential firing (Fig. 9). We quantified the average frequency of action potentials before and after addition of L-NOARG (Fig. 9d). A one-sided t-test was used to test the hypothesis that blockade of nitric oxide synthase would cause a greater increase in action potential firing aboral to a propulsive contraction (recorded in the distal colon) than proximal; the $p$ values for test are found in Supplementary Table 4.

The boxplots and the t-tests show that in the proximal colon site, there is no significant change in the number of action potentials, following the application of L-NOARG prior to and during propulsive contractions, (Fig. 9; $N = 4$). In contrast, for the distal colon, L-NOARG induced a significant increase in the total number of action potentials. Comparison of recordings in the presence and absence of L-NOARG revealed it was possible to identify the period during which distal EJPs are inhibited by nitric oxide release. This is shown by the black bars in Fig. 9b. After addition of L-NOARG these timepoints show more continuous action potential firing (Fig. 9b, c).

**Is uniform distension of the colon required for temporal coordination of ENS activity along the colon?** It was of particular interest to identify whether uniform distension along the colon was a prerequisite for synchronization of ENS activity and the generation of spatially and temporally coordinated EJPs over long distances. In 8 out of 12 occasions ($N = 4$), when the colon was distended with fluid, it was noted that the propulsive contraction incompletely expelled fluid from the colon (see Supplementary Fig.3A). Under these conditions, approximately half the colon remained contracted, whilst the remaining segment of colon remained distended with fluid (see Supplementary Fig.3A). Under these conditions, when the electrodes were separated by 30 mm, it was evident that a repetitive discharge of EJPs still became temporally coordinated between the proximal and distal colon (Supplementary Fig.3B and Supplementary Movie 5 and 6). This revealed that uniform colonic distension was not a prerequisite for long-range temporal coordination of EJPs (Supplementary Fig.3). See an example in real time of this activity

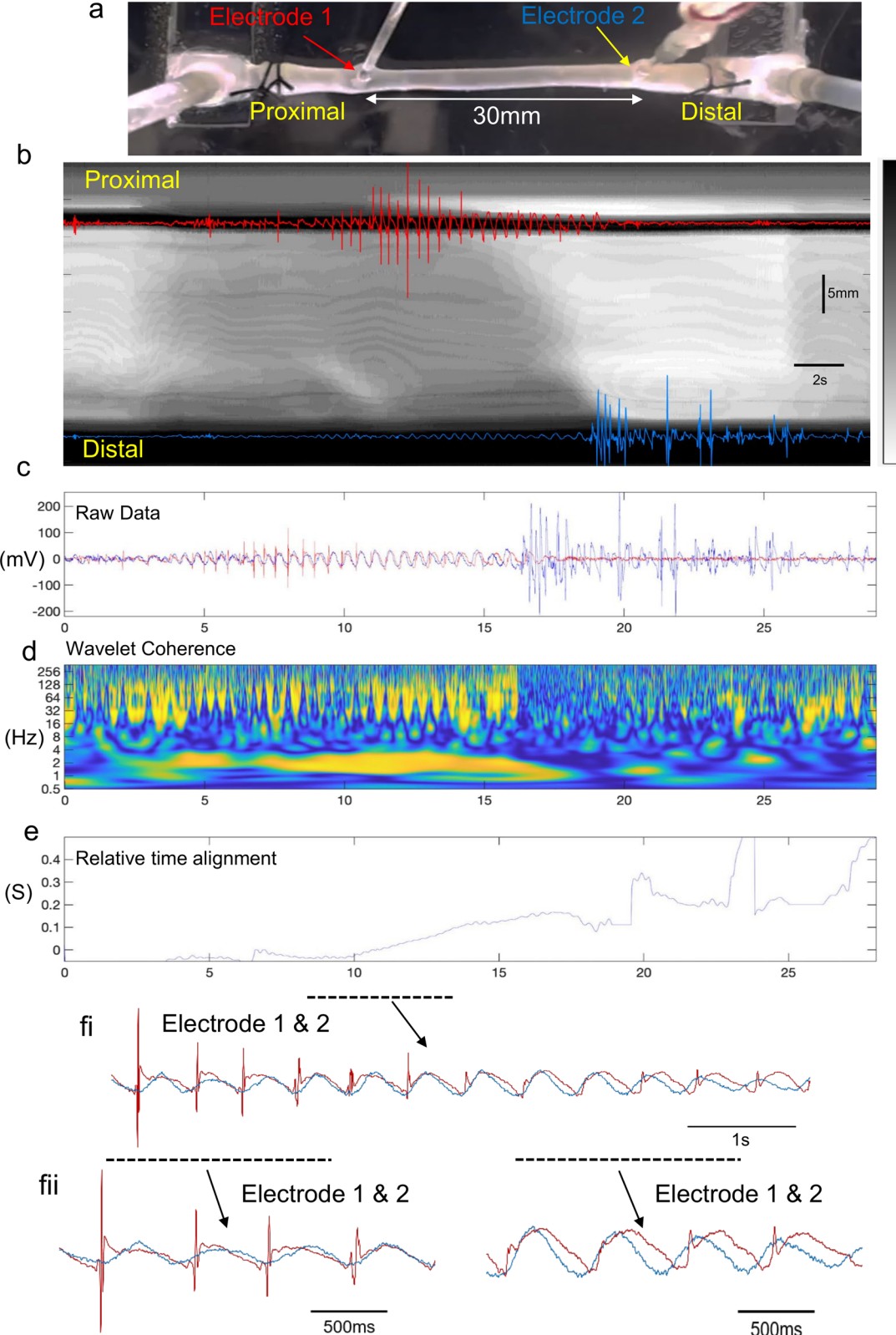

from 9 to 29 seconds in Supplementary Movie 5 and on expanded time scale in Supplementary Movie 6.

In a separate cohort of experiments, it was noted in three mice, that temporally coordinated EJPs could discharge over 30 mm length of colon without uniform distension of the colon, and without propulsion taking place, despite maintained (tonic) contraction of part of the length of colon (see Supplementary Fig.4 & Fig.5 and Supplementary Movie 7). All data and movie files from this study are freely available at ref. [50].

## Discussion

It is well known that isolated segments of intestine can generate propagating contractions of smooth muscle[17,20]. But, how the different neurochemical classes of neurons in the ENS are

**Fig. 3 Simultaneous video imaging colonic wall movements with electrical recordings made from two independent sites along the smooth muscle.** **a** shows a photomicrograph of the isolated colon and location of two independent extracellular recording electrodes, separated by 30 mm, where one electrode is located in the proximal/mid colon, the other in the distal colon. **b** Spatio-temporal map with superimposed simultaneous electrical recordings from the mid colon. The white band shows the propagating contraction associated with propulsion of fluid. **c** Superimposed electrical recordings from the proximal and distal colon. **d** Wavelet coherence of electrical activities between electrical recordings from electrode 1 (proximal colon) and electrode 2 (distal colon). The yellow band shown from ~6 to 17 s shows ~2 Hz peak coherence. **e** The relative time alignment between electrical recordings shown in panel **c** are presented. There is a small temporal difference between EJPs until ~17 s where the time difference increases and the propulsive wavefront passes. **fi** shows an expanded period represented by the black bar from panel **c**–**e**, where the close temporal synchronization between recordings EJPs is shown. **fii**, shows two different periods of recordings (represented by the two black bars) take from panel **fi**. The recording on the left hand side shows close synchronicity between EJPs, while the right hand panel recording in **fii** shows EJPs are phase shifted slightly by ~100–200 ms.

temporally activated along the colon, to cause propagating neurogenic contractions of the smooth muscle cells has remained uncertain. In this study, we took advantage of two recent advances; one in which it was possible to record electrical activity from the mouse colonic smooth muscle during dynamic spatio-temporal mapping, to infer firing patterns of different neurochemical types of neurons in the ENS[27]; and the other, the ability to record simultaneously propulsive movements and the underlying smooth muscle electrical activity generated by enteric neural circuits[44].

The findings here reveal how the different functional and neurochemical classes of neurons in the ENS are activated temporally and spatially along the length of the gut, to generate a mechanically evoked propagating neurogenic contraction of the smooth muscle resulting in propulsion of liquids. We reveal that large assemblies of ascending and descending interneurons fire in a coordinated and repetitive fashion at ~2 Hz to drive large populations of excitatory and inhibitory motor neurons at ~2 Hz, that lie both orally and aborally to a propagating neurogenic contraction. This is supported by the knowledge that individual motor neurons have projections less than 3 mm long and that individual inhibitory motor neurons have projections no more than 10 mm long and thus must be driven by ascending and descending interneuron circuits, which have projections up to 13 mm[51]. Exactly how the interneuronal circuits fire in a repetitive fashion is unclear, but it is apparent that they are interconnected to entrain bursting behaviour at ~1–2 Hz. Estimates suggest there are ~8000 nitrergic inhibitory neurons in the mouse colon and ~30,000 myenteric neurons in total[52]. We suggest that during propulsion, most, if not all of these myenteric neurons would likely be receiving coordinated bursts of fast synaptic inputs at ~1–2 Hz to generate EJPs in the muscle at a similar rate. Why the ENS of mouse colon has evolved to fire in coordinated and repetitive bursts at ~2 Hz is not clear, but this rate appears to be sufficient to cause a smooth muscle tetanus and sustained contraction facilitating propulsion in the distal region.

There has been a long-standing hypothesis that neurogenic propulsion of content along the bowel involves sequential activation of ascending excitatory nerve pathways orally and descending inhibitory nerve pathways aborally[16,17,20,21,53–57]. The findings here suggest more complex circuitry is activated, at least in the colon. The current findings show that during the aboral propulsion of fluid there is temporal coordination in firing of large populations of excitatory and inhibitory motor neurons, both orally and aborally to a propagating contraction of colonic smooth muscle (Fig. 10). This was represented by the finding that during propulsion of fluid, there is a repetitive discharge of cholinergic EJPs over large lengths of colon downstream (aboral) of the propagating contraction that become temporally coordinated with EJPs in the proximal colon at a time prior to when the contraction is initiated in the proximal colon (Supplementary Movie 1). Hence, there is ongoing coordinated neural activity in

the ENS over a long spatial range, even before any localized neurogenic contraction commences.

The data suggest that immediately prior to, and during the aboral propulsion of content, there is temporally coordinated activation of excitatory and inhibitory motor neurons, by a chain of shared interneurons. Importantly, despite coordinated activation of the ENS along the length of colon, colonic smooth muscle does not contract simultaneously during aborally propagating CMCs. The findings show this is because descending inhibitory pathways are strongly activated from the contractile region, leading to inhibition of the smooth muscle aborally. This is supported by the experiments using L-NOARG, which caused a large increase in EJP synchronization and action potential firing, in the region aboral to the contraction. Therefore, descending inhibitory pathways act to suppress downstream the excitation that is activated in parallel. This may correspond to the descending inhibition that was described by Bayliss and Starling in 1900[58], which may serve as a receptive relaxation, facilitating aboral propulsion of content. The effects of atropine, which abolished EJPs and revealed simultaneous IJPs, is also consistent with this mechanism. How the CMC contraction propagates into this area of inhibition is not clear.

Maintained colonic distension with a metal rod in the lumen also activated synchronized EJPs over considerable lengths of colon. When these EJPs were pharmacologically blocked, IJPs were revealed, albeit at a slightly lower frequency, suggesting that maintained distension also activates both inhibitory and excitatory motor neurons during this bursting behaviour, and that their bursts were both synchronized along more than 30 mm of colon.

It is known that the proximal region of colon displays neurogenic motor patterns[38,41,59], including unique enteric circuitry[38]. There are also clear neurochemical differences between myenteric neurons in the proximal and mid/distal colon[38]. However, our present current study shows that during distension and propulsion along the colon there is an intrinsic circuit that generates a ~2 Hz neural activity over a long range that must include temporal coordination of myenteric ganglia in the proximal and distal colon. Such neural activity can become temporally coordinated discharging EJPs and IJPs simultaneously during propulsive and non-propulsive CMC. An important observation was that when approximately half the colon remained in a tonic contraction (Supplementary Fig.3) while the other half of colon was under maintained distension by fluid, the ~2 Hz electrical pattern of EJPs still discharged in the smooth muscle in both regions (see, Supplementary Fig.3 and Supplementary Movie 5). Importantly, this means that neither uniform colonic distension nor propulsive movements were required to generate the ~2 Hz pattern of ENS firing (Supplementary Fig.4 and Fig.5 and Supplementary Movie 6 and Supplementary Movie 7). And, under maintained colonic distension with a metal rod in the lumen the ENS continued to generate cyclical bursts of coordinated neural activity every few minutes. These results uncover primordial circuity in the ENS, which when activated by a variety of stimuli and a

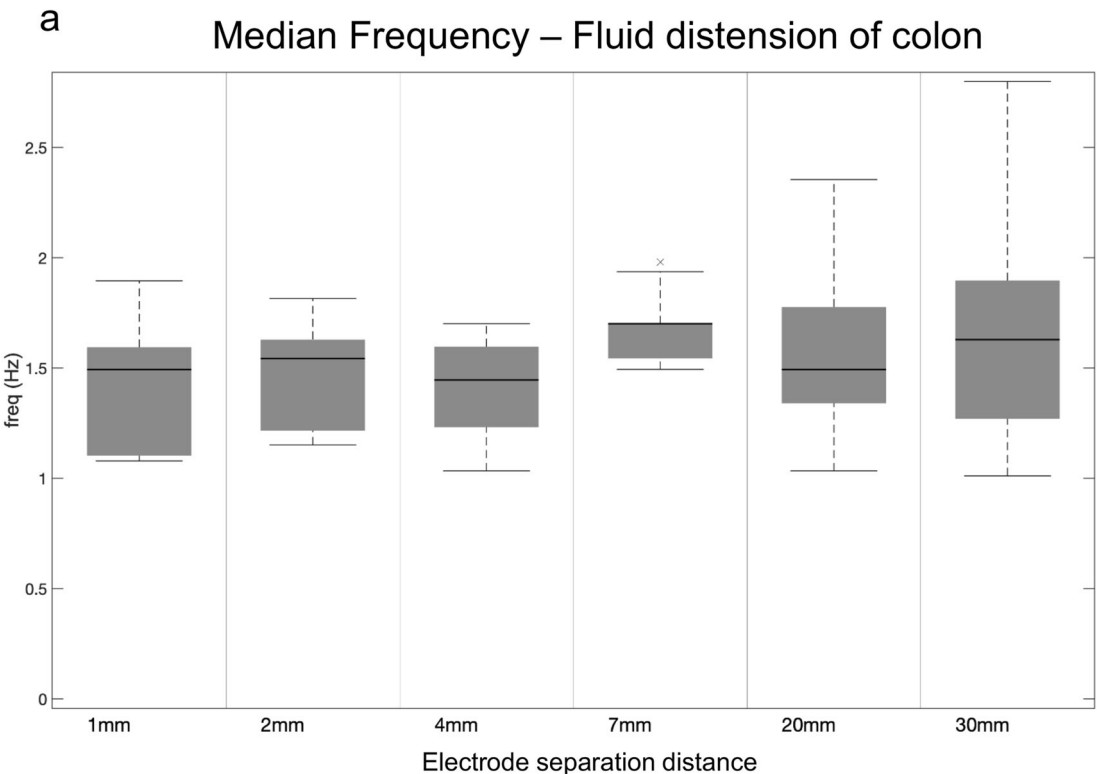

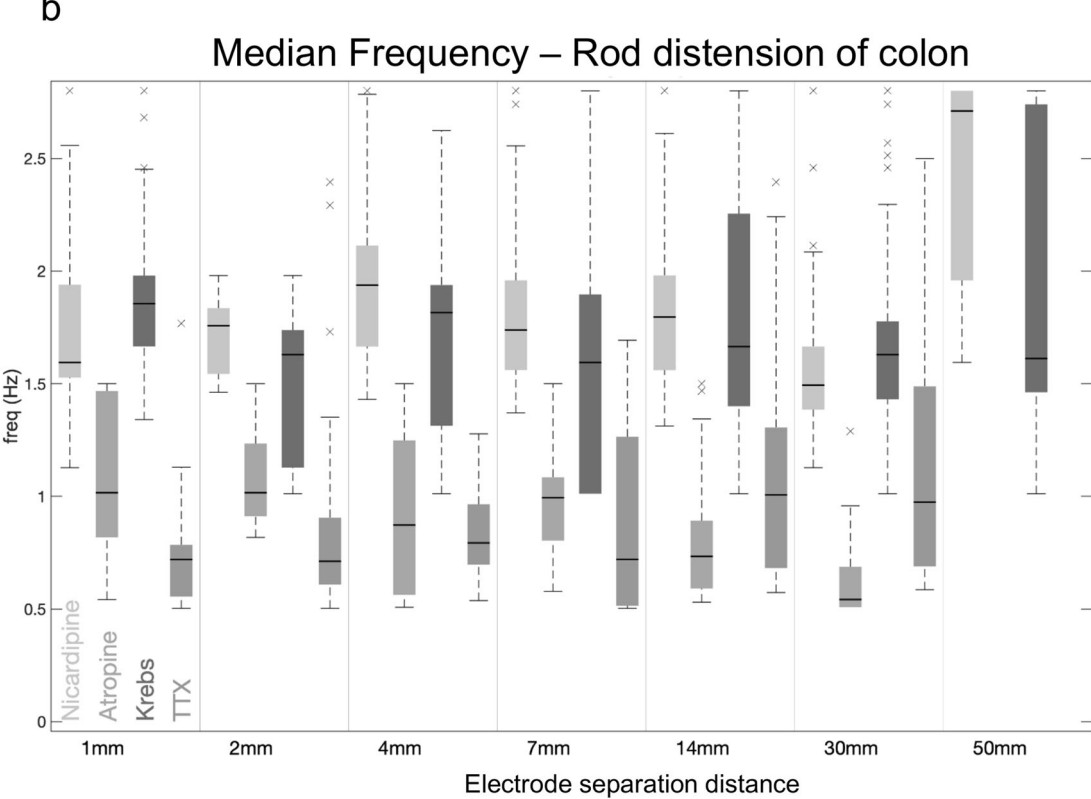

**Fig. 4 Median frequency of EJPs during propulsion elicited by fluid distension and rod distension. a** In response to fluid distension, boxplots show the distribution of the median frequency (during activity) as measured by the CWT, for the range of distances considered. **b** In response to rod distension, boxplots show the distribution of the median frequency (during activity) as measured by the CWT. In atropine, the coordinated IJPs occur at lower frequencies to the EJPs in nicardipine or normal Krebs solution.

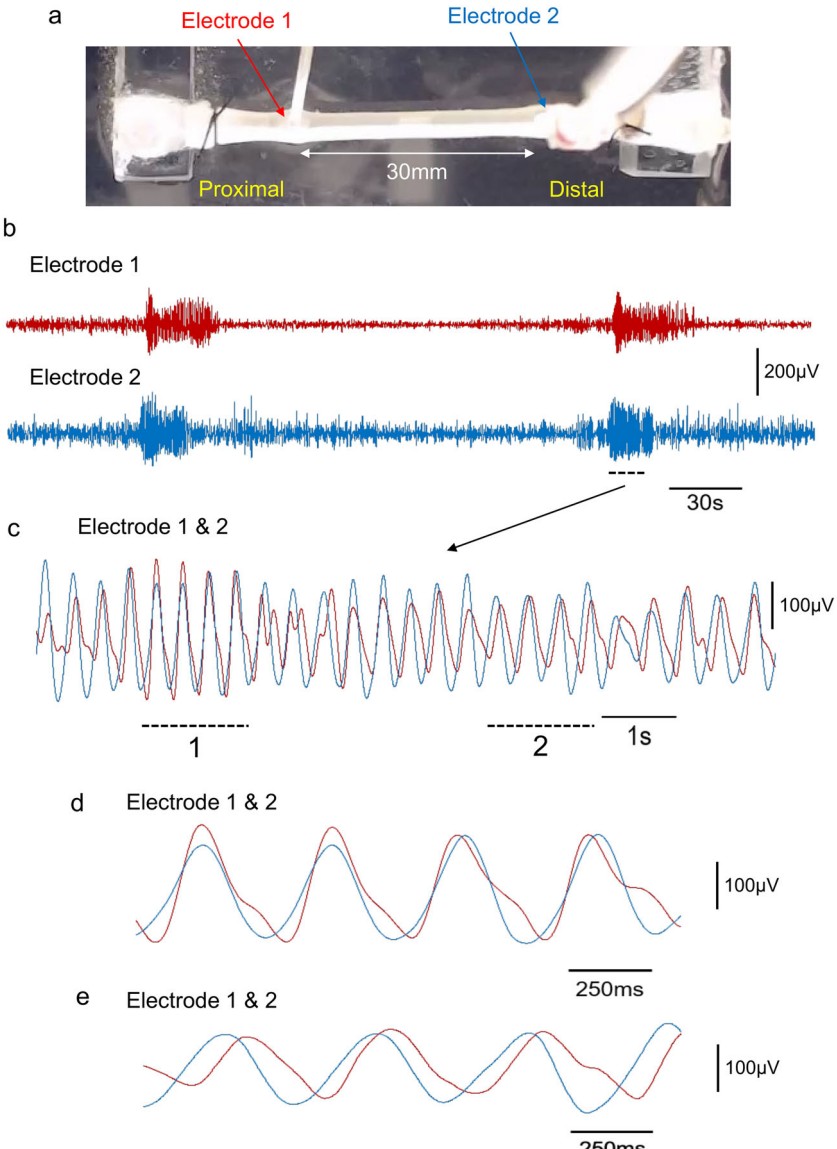

**Fig. 5 Coordinated electrical activities in colonic smooth muscle during maintained colonic distension with a metal rod of uniform diameter inserted through the lumen. a** Photomicrograph of the preparation of mouse colon with metal rod (inner diameter 2.41 mm through the lumen. The two electrodes are shown separated by 30 mm, one in the proximal-mid colon, the other in the distal colon. **b** shows simultaneous electrical recordings from the proximal and distal colon in the presence of nicardipine, where a coordinated burst of EJPs discharges at both electrodes. The periods represented by the bars labelled 1 and 2 are shown on expanded scale in **c** and **e**, respectively. **d** shows high temporal synchrony between EJPs, while a few seconds later in **e**, the EJPs are phase shifted by about 100 ms.

variety of experimental recording conditions (and independent of the mechanical states of the musculature) triggers a hard-wired neural pathway that discharges at ~2 Hz for ~20–30 seconds. Distension across the full length of colon was not a prerequisite for synchronization of the ENS circuits underlying propulsion. Furthermore, once elicited, the coordinated and repetitive activation of the ENS still maintains at firing rate of ~2 Hz, even though the colon can be partially contracted and lacks all movement of content. We have now revealed that a robust ENS circuit underlies both propulsive and non-propulsive behaviours (Fig. 10).

## Conclusion
The findings reveal the pattern of ENS activity that underlies the propulsion of fluid along the isolated colon. This mechanism involves coordinated firing of many thousands of ascending and descending interneurons that synaptically activate large populations of excitatory and inhibitory motor neurons, not only orally behind the bolus but also over considerable distances downstream, ahead of the propagating contraction wavefront (Fig. 10). Hence, the temporal delay in onset of smooth muscle contraction ahead of a propagating contraction is due to the preferential aboral projections of inhibitory motor neurons, which, when active, suppress the smooth muscle excitation by the concurrently active excitatory motor neurons. Our work reveals a unique feature of the ENS of the colon that underlies both non-propulsive and propulsive motor patterns. The study demonstrates the existence of large functional assemblies of neurons in the ENS that can self-organize to produce coordinated firing over many centimetres of colon, forming the basis of an important motor pattern in the colon.

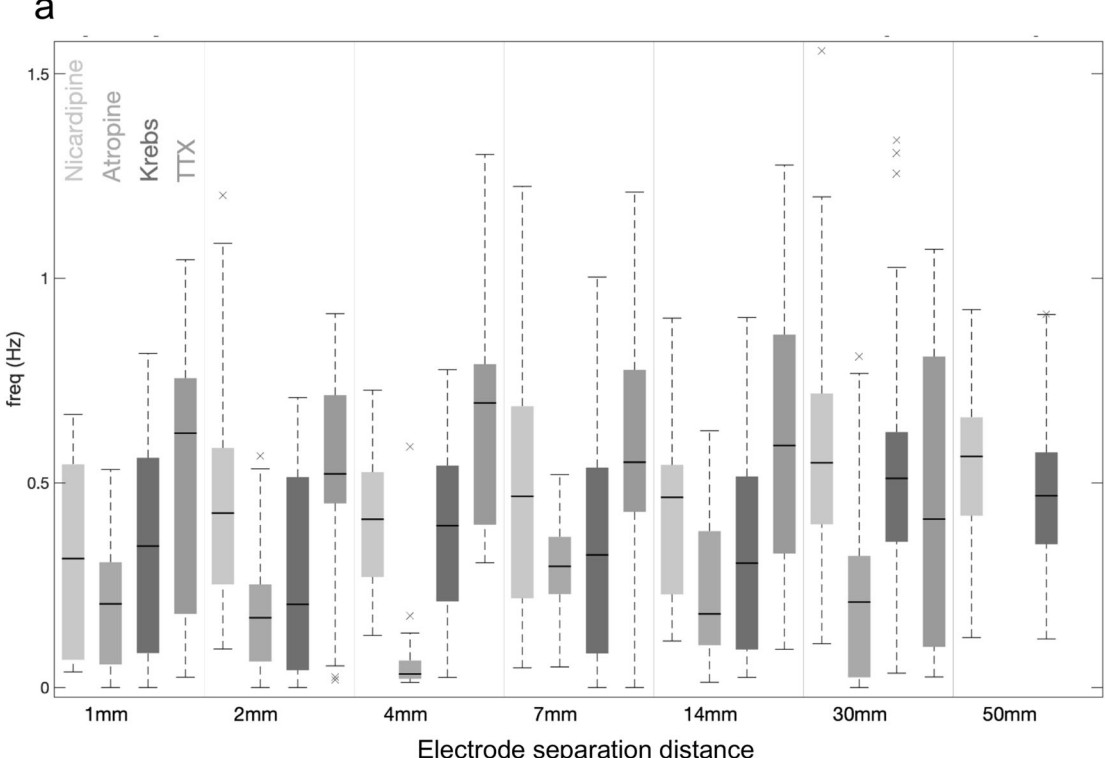

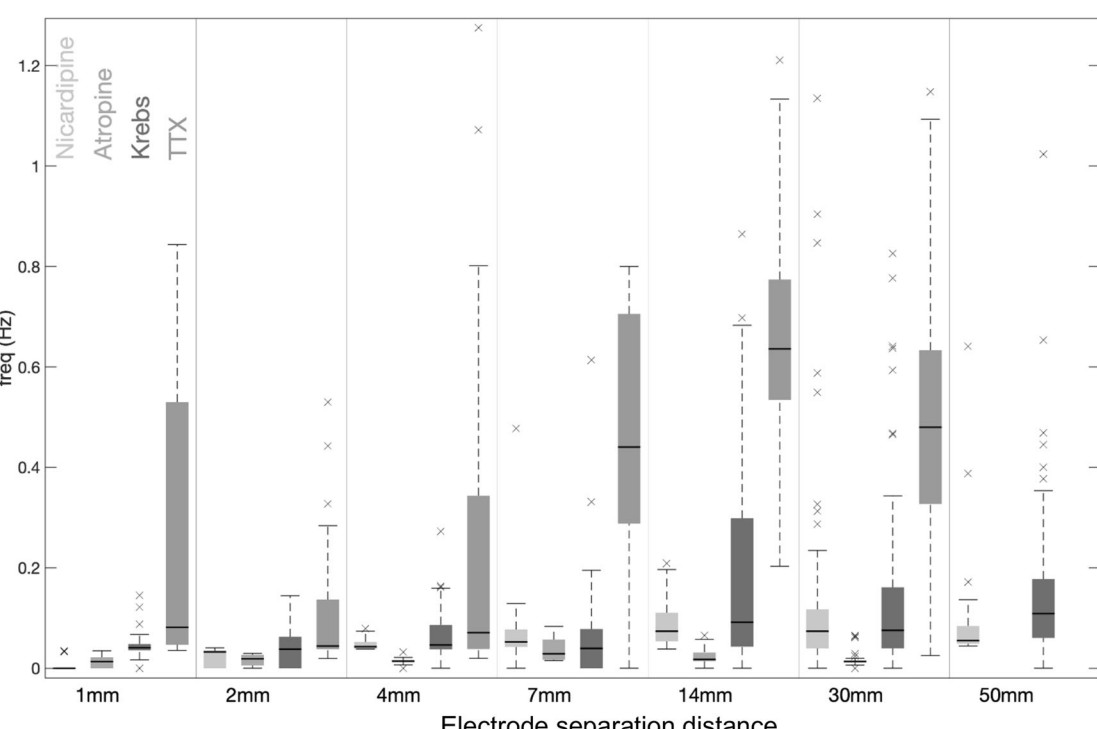

**Fig. 6 Summary plots showing the difference in frequency of junction potentials between two recording sites recorded over increasing separation distances along the colon.** Using a conventional boxplot, Plot **a** displays the distribution of the difference in frequency of junction potentials recorded prior to CMC electrical activity (measured between the proximal and distal colon). In TTX, there is a greater difference in frequency of junction potentials overall electrode separation distances. Plot **b** shows the frequency of junction potentials recorded from the proximal colon (i.e. frequency coordination) with junction potentials recorded more aborally, during periods of CMC electrical activity. In TTX, there is again a large difference in frequency of electrical activities overall electrode separation distances. The boxplots in **a** and **b** are conventional, where the horizontal dark line designates the median, the box edges the first and third quartiles, and the whiskers extending up to 1.5 times the interquartile range. See Methodology for Wavelet Coherence and Frequency Coordination Analysis for precise analytical details and Supplementary Table 2 for *t*-tests comparing medians.

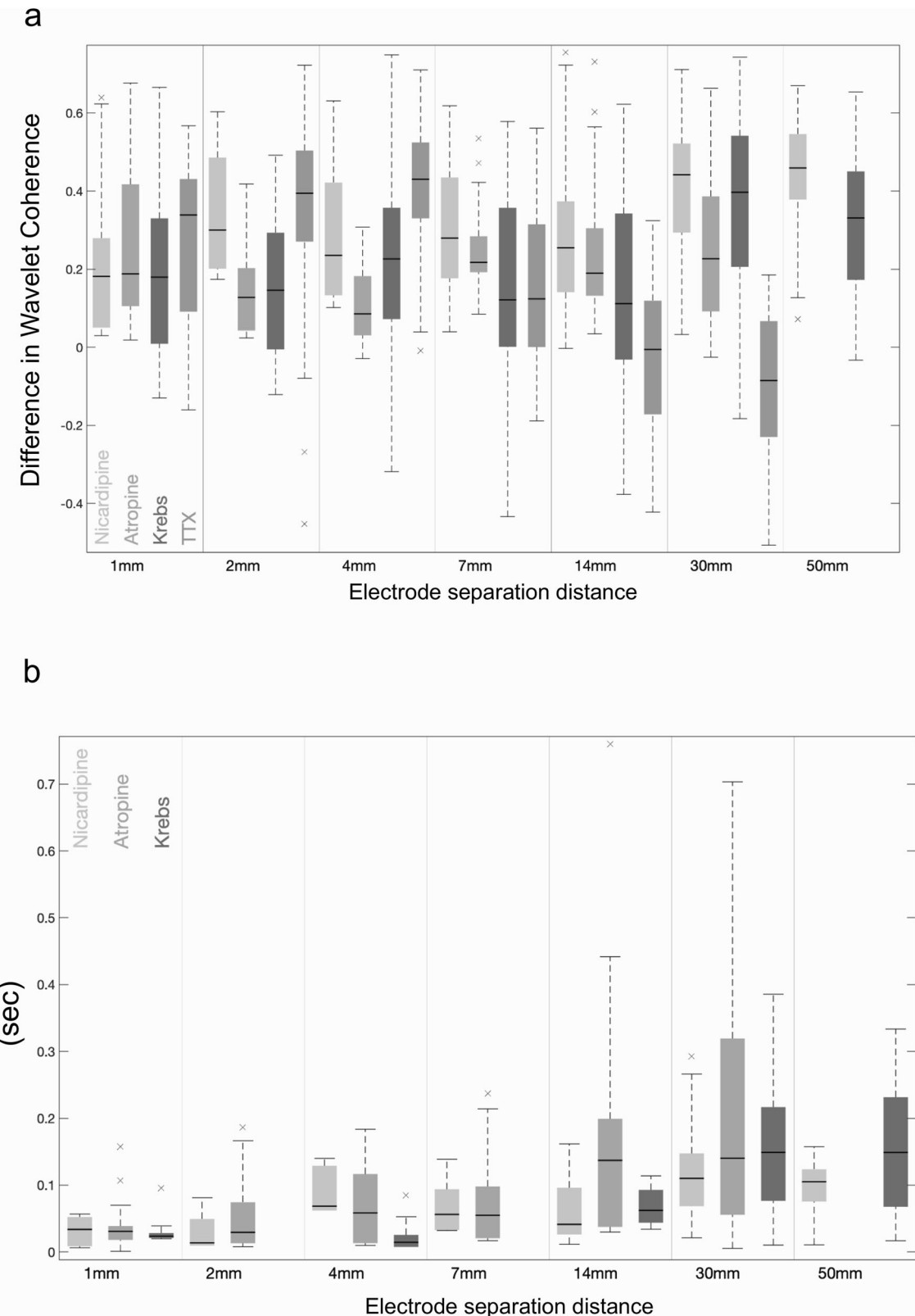

**Fig. 7 Summary plots showing the difference in wavelet coherence and the variation in time difference between junction potentials recorded during CMCs.** Plot **a** displays the difference in the median of the absolute value of WCOH during and before CMC propulsion, recorded between the proximal and distal colon. The distribution of the statistic is shown overall datasets displayed using a conventional boxplot; the solid dark line designates the median, the box edges the first and third quartiles, and the whiskers extending up to 1.5 times the interquartile range. Plot **b** shows the variation in time alignment between junction potentials recorded during CMCs, between the proximal and distal colon, over increasing electrode separations. As the distance between the electrodes increases, the degree of variability between the two recorded activities also increases. The time difference is estimated using the phase of the WCOH.

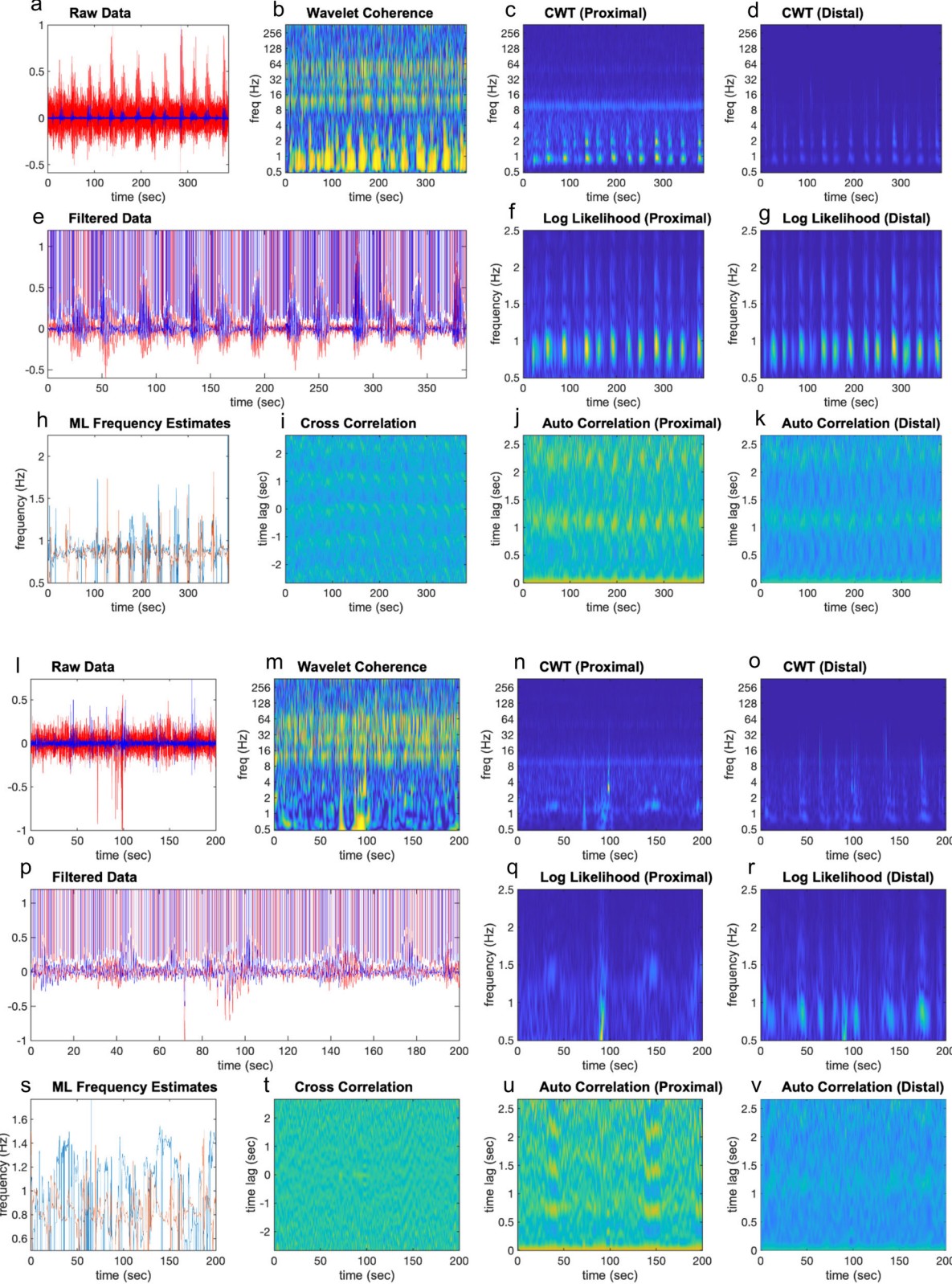

## Methods

**Tissue dissection.** C57BL/6 J mice of either sex (1–6 months of age) were euthanized using isoflurane inhalation overdose, followed by cervical dislocation, using a protocol approved by the Animal Welfare Committee of Flinders University (ethics approval no.861-13). A midline laparotomy was made and the entire colon, including the caecum and rectum was removed and placed into a Petri dish containing bubbled oxygenated Krebs solution. The organ bath set up used for video recording gut movements during propulsive motility consisted of a custom-made Perspex cavity (12 cm in length and 4 cm in width), where the isolated segment of colon (measuring ~6–7 cm in length) was positioned, similar to[27]. Videos were converted to diameter maps (DMaps) using custom-made software in Matlab (MathWorks, Natick, MA). Regions of minimal diameter (contraction) are

**Fig. 8 Effects of tetrodotoxin on spatial coordination of action potentials in colonic smooth muscle.** The plots within this figure have the same interpretation as those in Fig. 3. In this case, the plots pertain to the analysis of smooth muscle electrical activities during maintained colonic distension with a metal rod, over electrode separation distances from 2 mm and 30 mm. Plot **a**, **l** This plot contains the raw data for the proximal and distal colons, represented by the red and blue lines, respectively. Plot **b**, **m** This plot contains the WCOH (absolute value) between the proximal and distal colons. In contrast to Fig. 3, there is very little coherence at 2 Hz. For the 2 mm recording, there is coherence at ~0.8 Hz, whereas for the 30 mm recording, there is no coherence between 0-4 Hz. Plot **c**, **d**, **n**, **o** These plots contain the CWT (absolute value) for the proximal colon and distal colon respectively. In contrast to Fig. 3, the highest activity is between 0.8 and 1.2 Hz, not 2 Hz. Plot **e**, **p** The red and blue lines are traces of bandpass filtered data (filter limits 0.5 and 3.5 Hz) for the proximal and distal colon respectively. Plot **f**, **g**, **q**, **r** In both plots, a window (length 3 sec) has been slid over the bandpass filtered data in plots **e**, **p**. In each window instance, the likelihood that the data has a particular frequency is calculated; these plots display the log-likelihood. The frequency range for which the likelihood is calculated is given by the y-axis—in these examples from 0.5 to 2.5 Hz. Plot **h**, **s** These plots overlay the frequency maximizing the log-likelihood function for the proximal (plots **f**, **q**), and distal colon (plots **g**, **r**). For the 2 mm recording, the frequencies maximizing the likelihood functions are both ~0.8 Hz, whereas for the 30 mm recording, there is very little consistency in frequency. Plot **i**, **t**: In these plots, a window (length 3 sec) has been slid over the bandpass filtered data in plots **e**, **p**. In each window instance, the cross correlation between the proximal and distal colons is calculated, which these plots display. In contrast to Fig. 3, there are no periods of high or increased correlation, particularly for the 30 mm recording. Plot **j**, **k**, **u**, **v**: In these plots, a window (length 3 sec) has been slid over the bandpass filtered data in plot **e**, **p**. In each window instance, the auto-correlation function is calculated; plot **j**, **u** contains the auto-correlation function for the proximal colon, and plots **k**, **v** the auto-correlation function for the distal colon. In contrast to Fig. 3, the auto-correlation functions are not periodic at ~0.5 sec, rather, are weakly periodic between 0.8 and 1.25 sec, which is consistent with the frequency estimates displayed in plot **h**, **s**.

represented on DMaps as white pixels, whereas maximal diameter (dilatation) is represented by black pixels. Briefly, extracellular electrical recordings were made simultaneously with corresponding average changes in colonic wall diameter of selected regions in DMaps in Matlab. The action potentials were recorded by AC or DC amplifier. Data were exported to Microsoft Excel files. Frequencies were calculated from inter-spike intervals and averaged. Instantaneous frequency and force recorded from the proximal colon was also exported from LabChart for cross-correlation analysis using Matlab software. These data were down sampled from 1 to 0.1 kHz to reduce file size. The Krebs solution contained; (in mM concentrations: NaCl 118; KCl 4.7, $NaH_2PO_4$ 1; $NaHCO_3$ 25; $MgCl_2$ 1.2; D-Glucose 11; $CaCl_2$ 2.5; bubbled with 95% $O_2$ and 5% $CO_2$).

**Electrophysiological recordings.** We determined the intrinsic firing pattern of the ENS that underlies fluid propulsion, by quantifying the discharge rate of coordinated EJPs and IJPs in the smooth muscle. This was quantified over increasing distances along the colon, both during dynamic filling with fluid, and during propulsion. Custom-made flexible extracellular recording electrodes were used to monitor in real time the electrical activity of the smooth muscle at two sites along the length of colon. One electrode recorded in the most distal part of the proximal colon, while the second electrode was placed at variable distances to the first. Glass capillary suction electrodes (AgCl, 250 μm) were prepared using a heat-polished tip (0.58 mm internal diameter, 1 mm outer diameter; Harvard Apparatus) and applied to the serosal surface with a gentle suction. Electrical signals were acquired and amplified using an AC-coupled extracellular amplifier (ISO-80, WPI, Sarasota, FL, USA) with a 20 kHz low-pass filter using PowerLab 16/35, LabChart 8 (AD-Instruments, Castle Hill, NSW, Australia). In some experiments, we made simultaneous AC recordings (ISO-80 WPI) at the same time as DC recordings using a DAM-50 extracellular amplifier (WPI, Sarasota, Fl. USA). For DC recordings, a low-pass cut-off filter of 0.1 Hz and high-pass filter of 100 Hz were used. The ISO-80 primarily recorded fast action potentials (spikes); the DAM-50 recorded slower electrical events—largely excitatory and inhibitory junction potentials (EJPs and IJPs). Krebs solution was continuously superfused at a rate of ~5 ml/min at 35 °C.

**Drugs.** Nicardipine, atropine and $N_\omega$-Nitro-L-arginine methyl ester hydrochloride (L-NOARG) were obtained from Sigma Chemical Co. St. Louis, MO. USA and all made up in distilled water as stock solutions. Tetrodotoxin (TTX) was obtained from Alomone Laboratories. All drugs were prepared as stock solutions and kept refrigerated, then diluted to appropriate concentrations prior to use.

**Statistics and reproducibility.** For the *t*-tests performed, *p* values are calculated using the MATLAB function tcdf, which takes as its input the *t*-statistic and its degrees of freedom; tcdf outputs the corresponding *p* value. The wavelet analysis undertaken used the Matlab function wcoherence with the Mortlet wavelet. In the Supplementary Tables 1-4, the number of recordings (observations) that were made from the (N) number of preparations of colon that were studied is listed.

**Methodology for spike detection.** In the results section, the analysis is performed using in-house developed software in the MATLAB environment (MATLAB Version: 9.8.0.1380330 (R2020a) Update 2), for the Mac iOS (Mac OS X Version: 10.15.5). The analysis of the number of action potentials required a spike detection method. We utilized the approach described in ref. [60], in which a template is formed from the data, which is then correlated against the data. Spiking events are

then identified when correlation values exceed a threshold. The precise steps taken were:

(1) Apply a bandpass filter with filter limits 50 and 250 Hz (using the Matlab function "fir1" with default settings) to the raw data. Denote the filtered data by $\{y_n\}_{n=1}^N$.

(2) Create a template for the spiking events:

 a. Find index k maximizing $\{y_n\}_{n=1}^N$

 b. Define the template $\{z_n\}_{n=1}^{2L+1}$ by the following

$$z_n = y_{n+k-L}, n = 0, 1, \cdots, 2L + 1 \qquad (1)$$

(3) Correlate the template $\{z_n\}_{n=1}^{2L+1}$ against the data $\{y_n\}_{n=1}^N$; denote this correlation by $\{c_n\}_{n=1}^{N-2L}$.

(4) Define spiking events as those indices n such that

 a. $c_n > 3 \bullet s_c$

 b. $c_n > c_m$ for $m = n - M, \cdots, n - 1, n + 1, \cdots, n + M$

where

$$s_c = \sqrt{\sum_{n=1}^{N-2L} c_n^2 / (N - 2L)}. \qquad (2)$$

In our case, we found that a value of L = 15 was large enough to capture a spiking event. The purpose of M was to negate the effect of small spike-like waveforms that often accompanied a spiking event in our data; a value of M = 200 was sufficient.

**Methodology for wavelet coherence and frequency coordination analysis.** During propulsion events at the proximal colon, we observed the following two phenomena:

(1) An increase in the WCOH between the proximal and distal colon.

(2) Frequency Coordination between junction potentials recorded in the proximal and distal colon; that is the frequency maximizing many measures of frequency (CWT, log-likelihood, auto correlation—see Fig. 2) correlate closely.

In-house developed software in the MATLAB environment (MATLAB Version: 9.8.0.1380330 (R2020a) Update 2), for the Mac iOS (Mac OS X Version: 10.15.5) was used to systematically study the WCOH increase and frequency coordination of electrical activities. This software was applied to multiple recordings at various electrode separation distances (1–50 mm). For each recording, the propulsive contraction events were detected, and the following analytical protocol utilized:

We calculated the median of the WCOH before and during the propulsive contraction, where the WCOH was calculated at the frequency maximizing the CWT at the proximal colon; the difference in these medians is calculated at Step 4b in Supplementary Note 1. The Morlet wavelet was used for the WCOH and CWT. We found the frequency that maximized the CWT at the proximal and distal colon. We then calculated the median of the frequency deviation between the proximal and distal sites, before and during the propulsive contraction; the difference in these medians is calculated at Step 4a in Supplementary Note 1. The Morlet wavelet

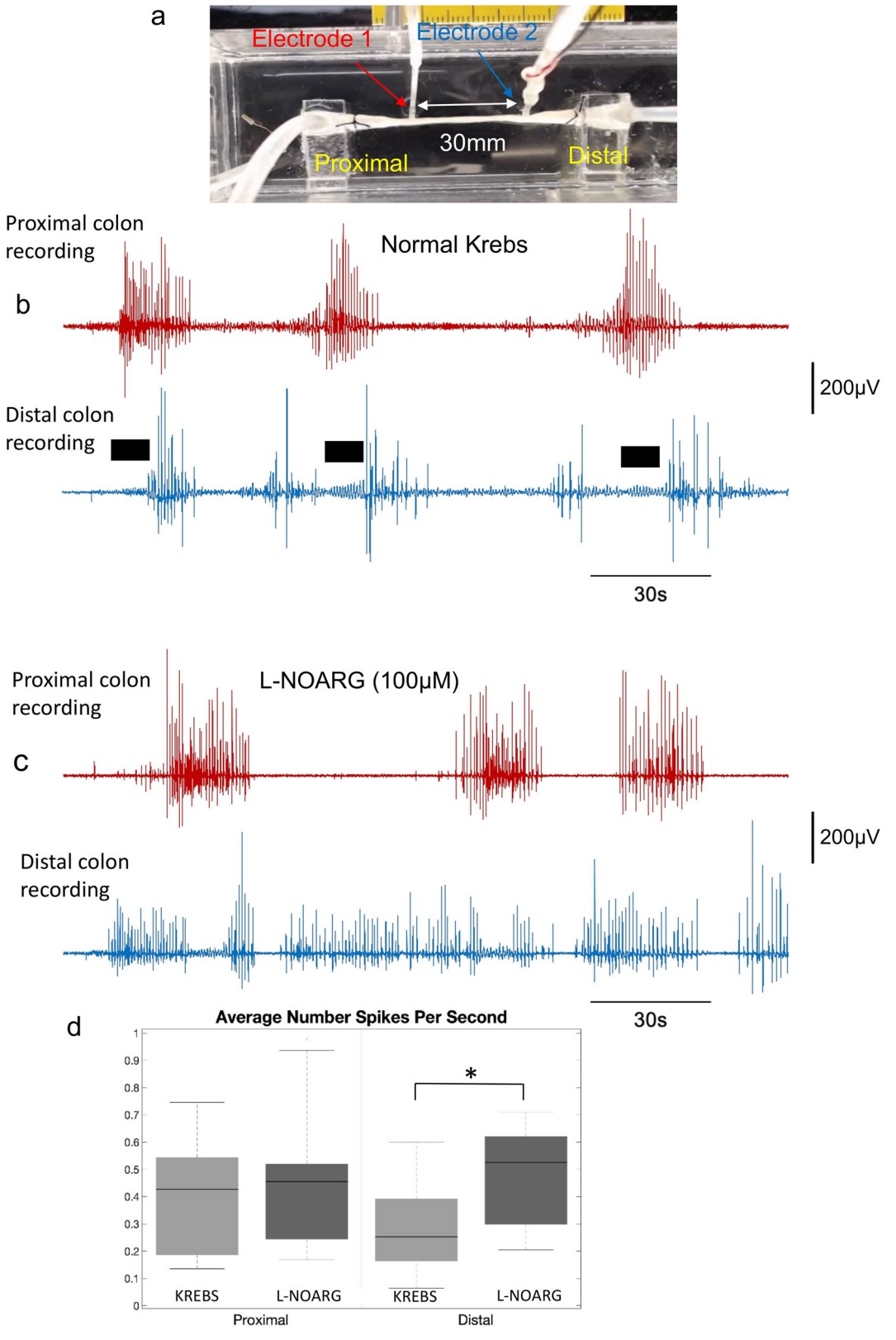

was used for the CWT. For both phenomena, a one-sided *t*-test was applied to difference in medians to assess the following:

Is there an increase in the (median) WCOH during a propulsive event?

Is there a decrease in the (median) frequency deviation during a propulsive event?

We also compared the WCOH and frequency coordination of electrical activities recorded in TTX to those recorded in the presence of Krebs, nicardipine

and atropine. More specifically, a one-sided, unpooled *t*-test was applied to difference in the medians to assess the following:

Is the (median) WCOH during activity for TTX lower than for Krebs, atropine and nicardipine?

Is the (median) frequency deviation during activity for TTX higher than for Krebs, atropine and nicardipine.

The Supplementary Note 1 describes the exact methodology to

**Fig. 9 Effects of acute blockade of nitric oxide synthesis on smooth muscle excitability during CMC propulsion. a** shows a schematic of the preparation used where recordings from the proximal and distal colon were made with electrodes separated by 30 mm. **b** shows a control recording in Krebs solution where fluid distension applied on three occasions elicits a discharge of action potentials first in the proximal colon, then with a temporal delay in the distal colon. The black bars indicate that when action potentials are discharging in the proximal colon there is a discharge of subthreshold EJPs occurring in the distal colon. **c** In the presence of L-NOARG to block nitric oxide synthesis, the temporal delay in action potential onset between the proximal and distal colon is abolished and there is a near tonic discharge in action potentials in the distal colon, compared to the proximal colon. **d** indicates the average number of spikes (action potentials) per second in both normal Krebs and L-NOARG, at both the proximal and distal regions of the colon. In the presence of L-NOARG, there is a significant increase in the average number of action potentials in the distal colon (see *) recording compared to proximal colon, but not in Krebs solution, indicating a preferential increase in excitation in the distal colon, following blockade of descending inhibition.

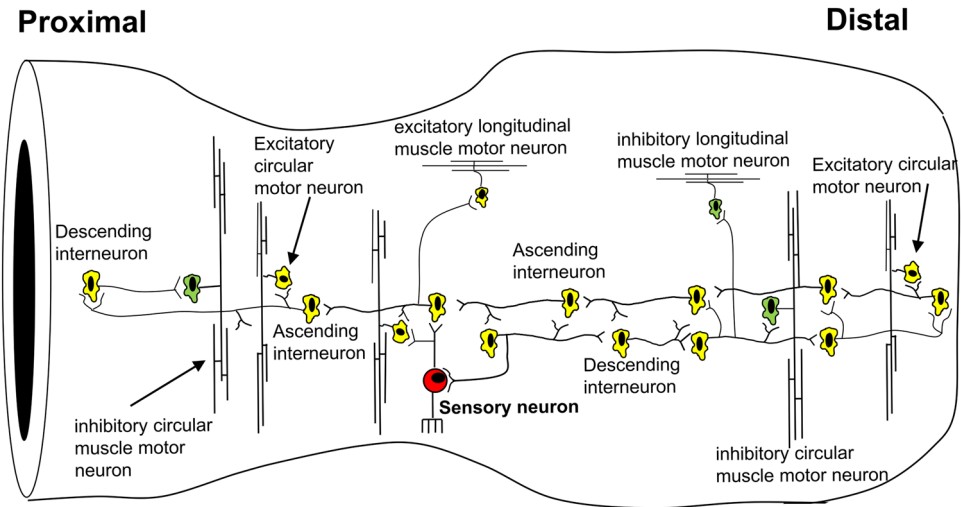

**Fig. 10 Intrinsic neural circuit identified that underlies propagating and non-propagating neurogenic contractions along the colon.** The central core of this enteric neural circuit is that ascending and descending interneurons synapse extensively with each other, such that when ascending or descending interneurons are activated this correspondingly activates descending or ascending interneurons, respectively. The major discovery is that long way downstream of any propagating contraction, there is repetitive activation of inhibitory and excitatory motor neurons at the same time. Aborally migrating contraction is temporally delayed downstream because activation of descending inhibitory motor neurons suppress EJPs from reaching action potential threshold by concurrently activated excitatory motor neurons. A major advance is that during the oral contraction in the proximal region, there is concurrent activation of excitatory and inhibitory motor neurons. The same circuit is found to occur in non-propagating contractions, see Supplementary Figs. 4&5 and Supplementary Movie 6.

Detect propulsion/activity at the proximal colon.

Measure the WCOH and frequency deviation before and during propulsion/activity at the proximal colon.

**Definitions used in Methods and Results**. Wavelet Coherence: measures the coherence between the proximal and distal colon. As the wavelet coherence is frequency dependent, and varies with time, we calculate the median of the wavelet coherence at the frequency maximizing the CWT at the proximal colon, before and during propulsion activity at the proximal colon. The resulting two medians were compared as a difference. Boxplots summarizing the difference in the two medians for nicardipine, atropine and Krebs data are given in Fig. 8a. These boxplots show that the level of coherence, as measured by the wavelet coherence is higher during a proximal propulsion/activity event than before propulsion.

Frequency coordination of electrical activities measure the degree to which the frequency of junction potentials in the proximal and distal colon (~1–2 Hz) correlate during propulsion/activity events at the proximal colon. The statistic used in the comparative studies was the median of the (absolute) difference in the frequencies maximizing the CWT (absolute value) at the proximal and distal colon during the period. Boxplots summarizing the statistics during and before propulsion/activity events at the proximal colon are given in Fig. 6a, b, respectively, for Nicardipine, Atropine and Krebs data. In these plots we see that the frequencies coordinate in time during a propulsive/activity event, whereas before the propulsion/activity event, the frequencies do not correlate.

Maximum Likelihood Frequency Estimation: these estimates, displayed in Fig. 2h, s, are the frequencies maximizing the log-likelihood function displayed in Fig. 2f, G and Fig. 2q, r, respectively. The log-likelihood estimates (calculated over sliding window) are calculated under the assumption that the signal observed within the window is a single sinusoid plus noise. This can be assumed as the data

have been bandpass filtered to remove higher frequency components. To compensate for the unknown phase, the log-likelihood is calculated for the analytic signal, which is the sum of the signal and it is Hilbert transform (where the Hilbert transform has also been multiplied by the sqrt of -1).

**Reporting summary**. Further information on research design is available in the Nature Research Reporting Summary linked to this article.

## Data availability
Data from this study are available at the git repository https://gitlab.com/jksorensen/long-range-synchronization-within-the-enteric-nervous-system[50].

## Code availability
The code in this manuscript can be accessed at the git repository https://gitlab.com/jksorensen/long-range-synchronization-within-the-enteric-nervous-system[61].

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

## Acknowledgements

Experiments in this study were supported by an NHMRC Project grant #1156416 to N.J.S. & Australian Research Council (ARC) grant #DP190103628 to N.J.S. We are deeply grateful to Dr. David Wattchow for his generous support to purchase extracellular amplifiers used in this study. We thank the SA Biomedical Engineering, Research and Teaching Team for supporting this project through the development and construction of heated bases, faraday cages and organ baths used in this study.

## Author contributions

N.J.S. designed, wrote, edited and submitted the paper. J.S. performed all the analysis, wrote the necessary code and made figures. L.T. performed all experiments. L.W. made the movies. M.C., S.J.B., P.D., T.H., H.H. and D.W. helped edit the paper.

## Competing interests

The authors declare no competing interests.
