## [Peer Review File · Communications Biology]

Reviewers' comments:

Reviewer #1 (Remarks to the Author):

Spencer et al. provide compelling evidence of a possible neural circuit in the enteric nervous system (ENS) driving enteric motility. They highlight key physiologic differences between the gastrointestinal tract relative to other hollow organs in terms of motility and fluid propulsion and hypothesize an important role for the ENS in driving these behaviors. The bulk of their evidence is based on a setup wherein ex vivo colonic tissue in an organ bath is simultaneously videoed during electrophysiology. Careful correlation of visual evidence of contraction and electrical recording of excitatory and inhibitory junction potentials (EJPs and IJPs) during fluid propulsion or constant distention with or without various chemical nerve blockers provide compelling evidence of spatially striated and temporally coordinated excitatory and inhibitory signals balanced to facilitate physiologic contractile activity. In particular the isolation of NOS dependent inhibitory signaling and disintegration of robust unidirectional propulsion in absence of these inhibitory signals supports their proposed model for ENS connectivity along the length of the colon driving the highly controlled colonic contractility. The findings are novel, significant, and interesting. Some of the observed phenomena, such as the significance and origin of the 2 Hz wavelet coherence, warrant further explanation.

Minor revision points:

1. Do the authors have any proposed explanations for what element(s) of the system may be producing the 2 Hz wavelet coherence and what significance it might have for colonic function? This should be added to discussion.
2. The provision of all data in Figures 5 and 7 is greatly appreciated. While inter-animal variability is to be expected, the amount of variability seems to be dramatically larger with larger inter-electrode distances, especially in the 30mm and 50mm conditions. Can the authors provide any details on why the data spread seems so dramatically different for these conditions, especially in Figure 5B? It would be relevant to know if there might be a fundamental limitation to the experimental setup or if the authors feel that this represents a physical limitation of the circuit itself based on the typical distance of axonal projections in the ENS and the interface between multiple units of a proposed feedback circuit along the length of the gut.
3. Given that the colonic content typically transitions from a fluid slurry to fairly solid pellets during a healthy bowel movement, have the authors considered repeating this study with solids which can be propelled (such as pellets or beads)? If so, are the outcomes consistent with the fluid propulsion studies?

Editorial;

1. Fig 5b title – “Frequency Difference between Junction Potentials over Distance during to Propulsion” – remove “to”
2. MATLAB/Mathworks citation in methods appears to be the only where the country is not indicated
3. Check methodology for Spike Detection – not sure if there was an accidental duplication or “...a template for the template is formed...” is accurate. Based on the detailed steps the former seems more likely.
4. The format for the table in the hypothesis testing section of the methods appears to be warped – it may be worthwhile to double check that it appears as intended.

Reviewer #2 (Remarks to the Author):

Spencer et al describe a newly developed technique to record excitatory junction potentials (EJPs) from colon tissue while colonic motor complexes (CMCs) are induced by fluid perfusion or colonic distension. They examined differences in both EJP and inhibitory junction potentials (IJP) activity by either placing the recording electrodes close together or up to 30 mm apart on the colonic preparation. Their main finding is that EJP and IJPs are synchronized with high coherence at 2Hz when recording from electrodes placed at distances of 1mm apart and 30mm apart. This suggests that the enteric nervous system network communication occurs over a longer range than previously expected. This has important implications for understanding mechanisms of colonic motility.

In summary, this manuscript reports scintillatingly interesting findings regarding the simultaneous activity of neurons distant to the incidence of distention in the colon, supporting synchronized long-range activity in colonic motility. It is true that the precise mechanisms of colonic motility remain unclear, and that unravelling these is very important. Overall, the findings from the current manuscript provide a new way of thinking about the biology of contractile patterns. To maximise on these findings, however, the manuscript could benefit from further clarification to better highlight these results and how they are building on previous research in the field of enteric neuroscience.

General Comments:

1. Can the authors comment on the speed of communication from the proximal to distal region of the colon correlate with previous works in this field?
2. Generally, when describing results in the main text, suggest a description of what was studied and the relevant result and a citation of the relevant figure at the end of the sentence. Currently, several of the sentences revolve around the figures themselves rather than reporting the results and then citing the relevant figure. E.g. in the second part of p6.
3. In the results section on p10, the comment that .."distension actually activates both inhibitory and excitatory motor neurons during this bursting behaviour...." seems to replicate current thinking about motor complexes (ie. Fig 2 in Fung and Vanden Berghe, 2020; PMID: 32424438), please clarify if this is the case and cite this recent review should this be considered relevant by the authors.
4. Please explain how the fluid was injected into the oral end of the colon preparation.

Comments by section:

Abstract:

1. Please articulate why the mechanism identified is "far more complex than expected"
2. It would be useful to mention the use of L-NOARG in this section since it is used as a major tool to assess the enteric neural circuitry in this study.
3. Please also mention the use of wavelet analysis and why this is important/what it revealed in the study.

Introduction

1. Page 3, Paragraph 2: Please expand the first occurrence of ENS abbreviation in main text.
2. Page 3, Paragraph 2.
Ref. 27 (Spencer, N. J. et al. Identification of a Rhythmic Firing Pattern in the Enteric Nervous System That Generates Rhythmic Electrical Activity in Smooth Muscle. *J Neurosci* 38, 5507-5522, doi:10.1523/JNEUROSCI.3489-17.2018 (2018).) refers to the discovery of a rhythmic firing pattern in the ENS that triggers smooth muscle electrical activity. This reference does not support this sentence regarding Hirschsprung's disease.
3. Page 3 – second last para – suggest reconsider sentence on Hirschsprung's disease – some patients do survive but need substantial medical assistance throughout life. Split into patients/animal models rather than rolling into one phrase as this is confusing (in an animal model it is a "model" of disease, whereas only patients can be diagnosed with the disease).
4. Page 4, Paragraph 2. Ref. 38 (Costa, M. et al. Neural motor complexes propagate continuously along the full length of mouse small intestine and colon. *Am J Physiol Gastrointest Liver Physiol* 318, G99-G108, doi:10.1152/ajpgi.00185.2019 (2020).) and 39 (Tan, W., Lee, G., Chen, J. H. & Huizinga, J. D. Relationships Between Distention-, Butyrate- and Pellet-Induced Stimulation of Peristalsis in the Mouse Colon. *Front Physiol* 11, 109, doi:10.3389/fphys.2020.00109 (2020).) refers to work in mouse and not other species.
5. P4 para 2 – remove hyphen between gastrointestinal and tract.

Results

1. Page 5, first paragraph: what is meant by "propagating contraction was initiated in the proximal colon at a sharp threshold"?
2. Page 5, Second paragraph: the whole figure 1 is referred to but an explanation of parts C-F would be beneficial.
3. Please clarify the significance of the wavelet analyses pseudocoloring of figures and how this

- presentation of data identifies pertinent findings. Explain why this approach was used.
4. Page 6, Paragraph 1. Does this refer to the gray scale background of Figure 2B?
 5. Page 6, Paragraph 2. This experiment and results were described but did not refer to any data shown. If data is shown in Supplementary Data, please refer accordingly.
 6. Page 5, final paragraph. Please rephrase so that the results are described with simple references to the Supplementary Movie. Ideally, a brief reference to Movie
 7. Page 6: Is the data shown for distal colon perfusion as there is no reference to it?
 8. Page 6 suggest modify final sentence of para 2 "distension evoked fluid propulsion appears always polarised..." meaning is unclear?
 9. Page 6 para 3: "...with varying electrode separations." Please provide a more precise description of this procedure (e.g. with varying electrode separation distances?).
 10. Page 8: While the inclusion of the null hypothesis explains your reasoning, this is not normally required in a research article. Suggest rephrasing the text and removing the reference to Hypotheses A & B, rather state in the text, precisely what was being tested, and why.
 11. Page 9: last paragraph please check if IJP has been defined.
 12. Page 9: the sentence "These correspond to cyclic motor complexes (CMCs) described in the introduction" is unnecessary. Remove or explain why the cyclical bursts correspond to CMCs.
 13. Page 10, first paragraph: Reference to figure 7A is vague and no mention of Figure 7b.
 14. Page 10, middle paragraph: appears out of place as refers to Figure 5.
 15. Page 10: please delete the following sentence "Using a metal rod, both hypotheses were accepted" and rewrite to describe the results, generally speaking an indepth interpretation is not required in the results section.
 16. Page 10: second para, sentence 3: rewrite to clarify if Table 2 refers to "Supplementary Table 2". Please also rewrite to emphasise the actual result, and refer to the table with the p values (rather than emphasizing p values initially). Overall this paragraph needs rewriting for clarification.
 17. Page 11: refers to table 2 but there is no table 2 included.
 18. Results, Page 11, Paragraph 2. "As" change to "a".
 19. Page 12: please refer to fig 9B when describing black bars not 9A. Also, in text please mention that this period depicted by black bars is also when fluid distension is occurring.
 20. Page 14: please amend final sentence "The effects of L-NOARG, a nitric oxide synthase blocker, suggest an explanation".
 21. Be consistent with formatting of measurement units (30 mm vs 30mm; the first is correct).

Discussion

1. Page 12, why is "MOVIE" in capital letters? (also in Fig 1 legend)?
2. Page 14, final para; data is plural, please amend.
3. Page 15, Paragraph 2. "2HZ" change to "2Hz".
4. Page 15, Paragraph 2. Consider rephrasing to "Importantly, it should be noted that...".
5. Page 15, final para; please rephrase this full sentence (due to grammatical errors) "We showed that regardless of how the colon was distended to elicit CMCs and even if only part of the length of colon...."

Methods

1. Overall, remove references to "Hypotheses A & B" and instead note what was being tested and why in plain language.
2. Tissue dissection section contains extra information not relevant to the dissection.
3. Drugs – please be consistent with noting catalogue numbers for each (or none) of the drugs. Currently a cat number is noted for hexamethonium but not others.
4. Drugs: "...All drugs were prepared as stock solutions "and" kept refrigerated..."
5. Statistics: check formatting for "for the MAC IOS" – should be Mac iOS?
6. Please explain what extra function was done by the "tinv" function, or else if this is a way to perform a t-test, just a simple statement is fine here.
7. More information is needed on what statistical test was performed to assess coherence.
8. Suggest moving the following sentence to results: "In our analysis we found that at ~2Hz, an increase in the CWT (cross wavelet transform) at the proximal colon was associated with an increase in the CWT at the distal colon. Moreover, the WCOH (wavelet coherence) increased, and the frequency of the oscillations..."
9. Please move the table (it is unlabeled) on page 20 to supplementary methods. Recommend formatting in a uniform font, remove grid format. Remove rationale sections and instead detail

each section separately eg "Calculation of Wavelet Decomposition". Remove reference to results and figures specific to the current study and instead report these in the results section of the manuscript together with the context of the experiments.

10. P17, para 1: please cite a relevant methods paper for the video recording of gut movements.

11. The statistics section should detail the statistical tests used, other information should be moved from this section. The final sentence should be in another subheading; eg: "Wavelet analysis" or "analysis of coherence" – it doesn't contain any statistical information; please move.

References:

1. Please remove the Von Haller 1755 reference, this is not necessary, suffice to note that it is well established in the main text that isolated segments can generate propagating contractions of smooth muscle. However, the precise mechanisms involving the enteric neural circuitry remain unclear.

Figures:

1. Figure 1 and 2 Fi and Fii: could benefit from a timeline added to the figure rather than referring to figure E and a box around the area which is magnified to make the figure clearer.

2. Figure 5B, Title of plot. "during to Propulsion" change to "during Propulsion".

3. Fig 6: please check if 200uV scale bar on IJP trace is correct as looks to be same size as fig 4b.

4. Figure 7, Legend. There are no red lines.

5. Figure 7. No data plot for Atropine at 50mm.

6. Figure 8: Suggest label y axis as percentage of events or CMCs

7. Figure 9D: suggest adding a line and star to indicate significant result of distal colon. Also, please indicate that black bars refer to fluid distension in the figure.

8. Figure 10: reword "The major discovery is that long way ..." also rephrase "is not because the neural activity hasn't reached..." (double negative and informal language). The final 2 sentences of this fig legend also need to be clarified.

9. Supplementary Figure 2, Legend. "F, shows the period represented by the dotted bar in C on expanded time..." change to "F, shows the period represented by the dotted bar in E on expanded time..."

Reviewer #3 (Remarks to the Author):

In this paper, Spencer and colleagues combined video imaging of colonic motility with smooth muscle electrophysiology recordings at two sites of varying distances to demonstrate synchronized rhythmic neurogenic motor activity along the length of the colon during fluid propulsion and maintained distension with a rod inserted into the lumen. This study follows on from their previous work demonstrating that large populations of enteric neurons fire in coordinated and repetitive bursts which generate rhythmic electrical activity in the smooth muscle.

1. In the introduction, it is mentioned that CMMCs originally described in mouse colon is present in other species. It would be helpful to actually refer to original work in other species as the ones cited (38, 39) refer to studies conducted in mouse.

2. In the results, sometimes the number of trials performed, or number of contractions assessed is unclear. For instance, for the p-values presented in tables 1 and 2 of the supplementary results, how many trials or how many contractions were examined for each distance of electrode separation and per condition?

3. It may be informative to show all the individual data points in the box plots in the figures.

4. It is described in the results that all spatial and temporal coordination of EJPs and IJPs was abolished by hexamethonium and tetrodotoxin. It would be interesting to also see this data presented.

5. Figure 6B shows electrical recordings in the presence of nicardipine + atropine. Can the authors

please clarify if the atropine data presented throughout the manuscript actually refer to recordings performed in the presence of both blockers or atropine alone?

6. On page 11, the summary of the L-NOARG analysis should refer to Table 3, not Table 2.

7. The authors reason that the delay in muscle contraction downstream of the advancing contraction may be explained by a concurrent activation of descending inhibitory nerve pathways over long ranges. To test this, L-NOARG was used to block the synthesis of the major inhibitory neurotransmitter nitric oxide and showed that this significantly increased the number of action potentials in the distal colon while the proximal colon was unaffected. Could the authors also show or comment on the effect of L-NOARG on the propagation of contractions and the temporal delay? Does the colon contract simultaneously in the absence of descending inhibition as predicted?

8. The final paragraph of the results describes a cohort of experiments where synchronized EJPs discharge along the whole colon, without uniform distension and without propulsion. In how many instances was this observed?

9. It is proposed in the discussion that "...immediately prior to, and during the aboral propulsion of content, there is simultaneous activation of excitatory and inhibitory motor neurons, by a shared chain of interneurons". However, it appears that during CMCs IJPs discharged at a lower frequency (~ 1Hz) compared to that of EJPs (~ 2 Hz). If the two populations of motor neurons are indeed driven by common inputs from shared interneurons as depicted in Figure 10, would it not be expected that both IJPs and EJPs ought to display a similar discharge frequency? If this interpretation is correct, then can the authors please provide an explanation for this disparity?

10. In the discussion, the following sentence needs rephrasing: "It is known that the proximal colon displays neurogenic motor patterns that are unique to the proximal colon, including unique enteric circuitry."

11. Might the title of Figure 10 ("Intrinsic neural circuit identified that underlies propagating neurogenic contractions along the colon") be reconsidered since the model proposed does not actually account for how this circuit may drive propagating contractions into a downstream region of active inhibition (as is also stated in the discussion on page 15)? Further, the "repetitive activation of inhibitory and excitatory motor neurons..." described is not uniquely observed during propulsive contractions but is also observed during non-propulsive contractions. How the coordinated and repetitive activation of this enteric circuit can give rise to both propagating and non-propagating motor patterns is unclear. Perhaps the authors can elaborate on this in the discussion.

12. Figure 8 shows that about 11% of events examined with electrodes attached were orally directed propagating contractions. Were these events also analyzed and if so, was synchronous electrical activity along the length of the colon also observed during these contractions?

13. In the methodology for spike detection section, the following sentence needs revision: "...in which a template for the template is formed..."

14. It would be helpful to show labels for the diameter scale in the DMaps in Figures 1B and 2B.

15. The legend for Figure 7 refers to a red line that is notably absent.

16. For reference it would also be good to have also some examples of the coherence, correlation and autocorrelation plots from the intestines that were treated with TTX and HEX. Furthermore it was mentioned that in HEX and TTX, synchrony was absent, but what happened to the signals? They were mostly gone I suppose ?

17. In the supplementary figures it is clear that there is a lot of variation between preparations. This is of course normal and to be expected. In this respect it would be informative to see some data on how consistent the measurements and advanced analysis are. Do repetitive stimuli generate very similar data and correlation plots? Is the variability mainly carried by repeats,

animals, and what happens over time, was there any influence of sex as both female and male mice were used. In this respect it should also be made very clear what $N = x$ actually means. Different measurements, different animals....

Reviewers' comments:

Thank you very much to Reviewer 1. Excellent comments were raised. We have included all suggestions and made all modifications.

Reviewer #1 (Remarks to the Author):

Spencer et al. provide compelling evidence of a possible neural circuit in the enteric nervous system (ENS) driving enteric motility. They highlight key physiologic differences between the gastrointestinal tract relative to other hollow organs in terms of motility and fluid propulsion and hypothesize an important role for the ENS in driving these behaviors. The bulk of their evidence is based on a setup wherein ex vivo colonic tissue in an organ bath is simultaneously videoed during electrophysiology. Careful correlation of visual evidence of contraction and electrical recording of excitatory and inhibitory junction potentials (EJPs and IJPs) during fluid propulsion or constant distention with or without various chemical nerve blockers provide compelling evidence of spatially striated and temporally coordinated excitatory and inhibitory signals balanced to facilitate physiologic contractile activity. In particular the isolation of NOS dependent inhibitory signalling and disintegration of robust unidirectional propulsion in absence of these inhibitory signals supports their proposed model for ENS connectivity along the length of the colon driving the highly controlled colonic contractility. The findings are novel, significant, and interesting. Some of the observed phenomena, such as the significance and origin of the 2 Hz wavelet coherence, warrant further explanation.

Minor revision points:

1. Do the authors have any proposed explanations for what element(s) of the system may be producing the 2 Hz wavelet coherence and what significance it might have for colonic function? This should be added to discussion.

Good question. We do not know why the ENS coordinates neuronal firing at ~2Hz. Our suspicion is that it may be just sufficient to evoke a smooth sustained contraction of the smooth muscle and able to be driven by bursts of fast EPSPs in from interneuronal pathways.

2. The provision of all data in Figures 5 and 7 is greatly appreciated. While inter-animal variability is to be expected, the amount of variability seems to be dramatically larger with larger inter-electrode distances, especially in the 30mm and 50mm conditions. Can the authors provide any details on why the data spread seems so dramatically different for these conditions, especially in Figure 5B? It would be relevant to know if there might be a fundamental limitation to the experimental setup or if the authors feel that this represents a physical limitation of the circuit itself based on the typical distance of axonal projections in the ENS and the interface between multiple units of a proposed feedback circuit along the length of the gut.

We appreciate these insightful questions. It is true, with increased electrode separation, there is decreased coordination between junction potentials. At 50mm separation, there is weaker spatial and temporal coordination between junction potentials. The original Figure 5B in the first submission is the now revised figure 6B in this resubmission. In this revised figure (Figure 6B), we added the new data with TTX, as requested. The results with TTX are striking in that paralyzing the ENS causes a major reduction in the spatial coordination of coordinated junction potentials (Fig.6B).

The working assumption is that over larger distances, there is a relative breakdown in the synchronization of interneuronal networks, leading to less synchronous EJPs over these distances.

3. Given that the colonic content typically transitions from a fluid slurry to fairly solid pellets during a healthy bowel movement, have the authors considered repeating this study with solids which can be propelled (such as pellets or beads)? If so, are the outcomes consistent with the fluid propulsion studies?

Good point. Yes, we have studied solid pellet propulsion. The 2Hz firing rate is preserved during the movement of single pellets (Movie in Spencer et al. 2018; J. Neurosci). The spatial field of synchronized EJPs is highly dependent upon the length of colon that is distended. With only a small area of distension (as with a single moving pellet), the area of muscle that shows synchronized EJPs is less. It was not possible to work this observation into the scope of the present study.

Editorial;

1. Fig 5b title – “Frequency Difference between Junction Potentials over Distance during to Propulsion” – remove “to”

Changed.

2. MATLAB/Mathworks citation in methods appears to be the only where the country is not indicated.

We have changed this description.

3. Check methodology for Spike Detection – not sure if there was an accidental duplication or “...a template for the template is formed...” is accurate. Based on the detailed steps the former seems more likely.

Thank you. We have corrected this.

4. The format for the table in the hypothesis testing section of the methods appears to be warped – it may be worthwhile to double check that it appears as intended.

Ok. Thanks. I believe the editorial staff will replot this.

Reviewer #2 (Remarks to the Author):

Thank you to Reviewer 2 for raising great comments. Very constructive. We have included all suggestions and made appropriate modifications.

Spencer et al describe a newly developed technique to record excitatory junction potentials (EJPs) from colon tissue while colonic motor complexes (CMCs) are induced by fluid perfusion or colonic distension. They examined differences in both EJP and inhibitory junction potentials (IJP) activity by either placing the recording electrodes close together or up to 30 mm apart on the colonic

preparation. Their main finding is that EJP and IJPs are synchronized with high coherence at 2Hz when recording from electrodes placed at distances of 1mm apart and 30mm apart. This suggests that the enteric nervous system network communication occurs over a longer range than previously expected. This has important implications for understanding mechanisms of colonic motility.

In summary, this manuscript reports scintillatingly interesting findings regarding the simultaneous activity of neurons distant to the incidence of distention in the colon, supporting synchronized long-range activity in colonic motility. It is true that the precise mechanisms of colonic motility remain unclear, and that unravelling these is very important. Overall, the findings from the current manuscript provide a new way of thinking about the biology of contractile patterns. To maximise on these findings, however, the manuscript could benefit from further clarification to better highlight these results and how they are building on previous research in the field of enteric neuroscience.

General Comments:

1. Can the authors comment on the speed of communication from the proximal to distal region of the colon correlate with previous works in this field?

We believe the reviewer is referring to the speed of communication with regards to the communication of the ENS network ? In this regard, we showed in Spencer N et al. 2005; J. Physiol; that EJPs synchronize over a spatial field of around 7mm in preparations that are maintained (pinned) under circumferential stretch and not given any recover period (ie. no recovery from stretch). In contrast, when isolated tube preparations of colon are allowed to recover after an acute distension (with fluid), we found in this study (acute fluid distension, the degree of coordination and spatial synchronisation was considerably larger.

2. Generally, when describing results in the main text, suggest a description of what was studied and the relevant result and a citation of the relevant figure at the end of the sentence. Currently, several of the sentences revolve around the figures themselves rather than reporting the results and then citing the relevant figure. E.g. in the second part of p6.

Thanks for pointing this out. We have reduced discussion of individual results and figures. We have modified sentences of this nature to be more helpful to readers and followed this suggestion to speak more of the collective data and then refer to individual figures. Much of the results section has been tightened and reduced to address this. Thanks.

3. In the results section on p10, the comment that .."distension actually activates both inhibitory and excitatory motor neurons during this bursting behaviour...." seems to replicate current thinking about motor complexes (ie. Fig 2 in Fung and Vanden Berghe, 2020; PMID: 32424438), please clarify if this is the case and cite this recent review should this be considered relevant by the authors.

We have quoted the nice review by Fung et al. We had overlooked the similar diagram. This circuit was first proposed and published in 2005, in Spencer et al. J. Physiol. for stationary (maintained distension) reflex pathways.

4. Please explain how the fluid was injected into the oral end of the colon preparation.

We have added a sentence in the methods about this.

Comments by section:

Abstract:

1. Please articulate why the mechanism identified is “far more complex than expected”

We did not expect such a large area of colon to demonstrate such temporally coordinated neural activity. We have modified this sentence by deleting the word “far”.

2. It would be useful to mention the use of L-NOARG in this section since it is used as a major tool to assess the enteric neural circuitry in this study.

We would love to, but word limits for the abstract precluded this.

3. Please also mention the use of wavelet analysis and why this is important/what it revealed in the study.

Ok. This has been included in the methods section.

Introduction

1. Page 3, Paragraph 2: Please expand the first occurrence of ENS abbreviation in main text.

Done.

2. Page 3, Paragraph 2.

Ref. 27 (Spencer, N. J. et al. Identification of a Rhythmic Firing Pattern in the Enteric Nervous System That Generates Rhythmic Electrical Activity in Smooth Muscle. *J Neurosci* 38, 5507-5522, doi:10.1523/JNEUROSCI.3489-17.2018 (2018).) refers to the discovery of a rhythmic firing pattern in the ENS that triggers smooth muscle electrical activity. This reference does not support this sentence regarding Hirschsprung's disease.

True, this was the wrong reference. We have replaced it with the more appropriate reference to Ro S et al. 2006.

3. Page 3 – second last para – suggest reconsider sentence on hirschsprung’s disease – some patients die but need substantial medical assistance throughout life. Split into patients/animal models rather than rolling into one phrase as this is confusing (in an animal model it is a “model” of disease, whereas only patients can be diagnosed with the disease).

Good point. Done.

4. Page 4, Paragraph 2. Ref. 38 (Costa, M. et al. Neural motor complexes propagate continuously along the full length of mouse small intestine and colon. *Am J Physiol Gastrointest Liver Physiol* 318, G99-G108, doi:10.1152/ajpgi.00185.2019 (2020).) and 39 (Tan, W., Lee, G., Chen, J. H. & Huizinga, J. D. Relationships Between Distention-, Butyrate- and Pellet-Induced Stimulation of Peristalsis in the Mouse Colon. *Front Physiol* 11, 109, doi:10.3389/fphys.2020.00109 (2020).) refers to work in mouse and not other species.

We have modified this reference position.

5. P4 para 2 – remove hyphen between gastrointestinal and tract.

Thanks for spotting this. Done.

Results

1. Page 5, first paragraph: what is meant by “propagating contraction was initiated in the proximal colon at a sharp threshold”?

We have modified this statement to say “propagating contraction was initiated in the proximal colon.”

2. Page 5, Second paragraph: the whole figure 1 is referred to but an explanation of parts C-F would be beneficial.

We have modified this paragraph to encompass Figure 1 all panels.

3. Please clarify the significance of the wavelet analyses pseudocoloring of figures and how this presentation of data identifies pertinent findings. Explain why this approach was used.

The colouring represents intensity, the lowest value being blue and the highest yellow. The yellow band exemplifies the greatest frequencies of synchronization. The distinct band at 2Hz shows this occurs at close electrode separations, but during periods out of the recording the synchronization is reduced. This is shown by the reduced extent of the thick yellow band.

4. Page 6, Paragraph 1. Does this refer to the gray scale background of Figure 2B?

Yes. The Figure 2B refers to the movie 2B. We have stated this now.

5. Page 6, Paragraph 2. This experiment and results were described but did not refer to any data shown. If data is shown in Supplementary Data, please refer accordingly.

The entire paragraph that the reviewer refers to has been completely rewritten. It is much more succinct now. The data that the reviewer suggested we refer to is now clear in the Tables 1-4. These revised tables now show how many preparations and how many animals the wavelet coherence was determined by. The original submission Figure 3 has now become Figure 2. We have added a new Table, Supplementary Table 4 for the requested L-NOARG data. And the TTX data (requested) has been included in revised Supplementary Tables 2 and 3 (which also state how many recordings from how many animals were used).

6. Page 5, final paragraph. Please rephrase so that the results are described with simple references to the Supplementary Movie. Ideally, a brief reference to Movie

Good point. We deleted the whole paragraph and tightened up.

7. Page 6: Is the data shown for distal colon perfusion as there is no reference to it?

We have deleted the reference to distal colon infusion of fluid. It didn't really seem necessary.

8. Page 6 suggest modify final sentence of para 2 “distension evoked fluid propulsion appears always polarised...” meaning is unclear?

As mentioned above, this paragraph and distal colon infusions is deleted.

9. Page 6 para 3: ...with varying electrode separations.” Please provide a more precise description of this procedure (e.g. with varying electrode separation distances?).

This has been deleted and replaced with clearer wording

10. Page 8: While the inclusion of the null hypothesis explains your reasoning, this is not normally required in a research article. Suggest rephrasing the text and removing the reference to Hypotheses A & B, rather state in the text, precisely what was being tested, and why.

All reviewers didn't like our statements of null hypotheses. We deleted them.

11. Page 9: last paragraph please check if IJP has been defined.

It has now. Thanks.

12. Page 9: the sentence “These correspond to cyclic motor complexes (CMCs) described in the introduction” is unnecessary. Remove or explain why the cyclical bursts correspond to CMCs.

We have deleted this sentence.

13. Page 10, first paragraph: Reference to figure 7A is vague and no mention of Figure 7b.

We have corrected the order of these figures. Thanks.

14. Page 10, middle paragraph: appears out of place as refers to Figure 5.

True. This has been deleted and whole paragraph reworded.

15. Page 10: please delete the following sentence “Using a metal rod, both hypotheses were accepted” and rewrite to describe the results, generally speaking an in depth interpretation is not required in the results section.

We have deleted reference to accepting the hypotheses.

16. Page 10: second para, sentence 3: rewrite to clarify if Table 2 refers to “Supplementary Table 2”. Please also rewrite to emphasise the actual result, and refer to the table with the p values (rather than emphasizing p values initially). Overall this paragraph needs rewriting for clarification.

This whole paragraph has been rewritten to appropriately refer to the Supplementary Tables. As requested by a reviewer, the methodology Table that described the code to determine wavelet coherence (originally on page 20) has been moved to supplementary Table 5. Also, as

recommended by the reviewers, we have deleted the null hypotheses. P values included in the revised Tables, as requested.

17. Page 11: refers to table 2 but there is no table 2 included.

The original supplementary tables are now tables in the results.

18. Results, Page 11, Paragraph 2. "As" change to "a".

Corrected.

19. Page 12: please refer to fig 9B when describing black bars not 9A. Also, in text please mention that this period depicted by black bars is also when fluid distension is occurring.

We have clarified in legend to new Figure 9 that in fact, the black bar was to indicate where EJPs could be recorded in the distal recording electrode whilst the proximal recording electrode has action potentials. These EJPs in the distal recording electrode can be seen to be suppressed from reaching action potential threshold, but in C, when L-NOARG is present. There are action potentials occur at the distal electrode. This is now explained clearer in the text that what the black bar refers to in B. Thanks for clarifying this.

20. Page 14: please amend final sentence "The effects of L-NOARG, a nitric oxide synthase blocker, suggest an explanation".

Changed now.

21. Be consistent with formatting of measurement units (30 mm vs 30mm; the first is correct).

Ok. Corrected.

Discussion

1. Page 12, why is "MOVIE" in capital letters? (also in Fig 1 legend)?

We have corrected this to Movie.

2. Page 14, final para; data is plural, please amend.

Good point. Corrected.

3. Page 15, Paragraph 2. "2HZ" change to "2Hz".

Corrected.

4. Page 15, Paragraph 2. Consider rephrasing to "Importantly, it should be noted that...".

Corrected.

5. Page 15, final para; please rephrase this full sentence (due to grammatical errors) "We showed

that regardless of how the colon was distended to elicit CMCs and even if only part of the length of colon....

Corrected.

Methods

1. Overall, remove references to “Hypotheses A & B” and instead note what was being tested and why in plain language.

We have deleted reference to Hypotheses A & B. And deleted the Null hypotheses.

2. Tissue dissection section contains extra information not relevant to the dissection.

3. Drugs – please be consistent with noting catalogue numbers for each (or none) of the drugs.

See below.

Currently a cat number is noted for hexamethonium but not others.

We have deleted Cat numbers, to be consistent.

4. Drugs: “....All drugs were prepared as stock solutions “and” kept refrigerated...”

Corrected. Thanks for noticing this.

5. Statistics: check formatting for “for the MAC IOS” – should be Mac iOS?

Corrected. Thanks.

6. Please explain what extra function was done by the “tinv” function, or else if this is a way to perform a t-test, just a simple statement is fine here.

We named the function incorrectly. Apologies. We in fact used the tcdf function, which calculates the p-value for a particular t-statistic with a certain number of degrees of freedom. We have added this to paper.

7. More information is needed on what statistical test was performed to assess coherence.

The quantity used to investigate the increase in the (wavelet) coherence before and during a propulsive contraction is given in Step 4(b) in the section *Methodology for Wavelet Coherence and Frequency Locking Analysis*. The quantity essentially compares the wavelet coherence before and during a propulsive contraction at the proximal site. More specifically, the median value of the wavelet coherence (at the frequency maximizing the continuous wavelet transform at the proximal site) is calculated before and during a propulsive contraction; box plots illustrating this median over all distances before and during the propulsive contraction are shown in Figure 6A and 6B respectively (Nicardipine, atropine and Krebs). Formal statistical tests for the increase in coherence were performed at each distance using a t-test were applied to quantity in Step 4(b); the p-val for these test are given in Supplementary Table 2 & 3 for Nicardipine, atropine and Krebs solution. In the methodology section we have added content explaining this.

8. Suggest moving the following sentence to results: “In our analysis we found that at ~2Hz, an increase in the CWT (cross wavelet transform) at the proximal colon was associated with an increase in the CWT at the distal colon. Moreover, the WCOH (wavelet coherence) increased, and the frequency of the oscillations...”

Done. Sentences have been moved.

9. Please move the table (it is unlabelled) on page 20 to supplementary methods. Recommend formatting in a uniform font, remove grid format. Remove rationale sections and instead detail each section separately eg “Calculation of Wavelet Decomposition”. Remove reference to results and figures specific to the current study and instead report these in the results section of the manuscript together with the context of the experiments.

We have moved this table. It is supplementary Table 5.

10. P17, para 1: please cite a relevant methods paper for the video recording of gut movements.

Now inserted relevant reference.

11. The statistics section should detail the statistical tests used, other information should be moved from this section. The final sentence should be in another subheading; eg: “Wavelet analysis” or “analysis of coherence” – it doesn’t contain any statistical information; please move.

We have modified it as suggested

References:

1. Please remove the Von Haller 1755 reference, this is not necessary, suffice to note that it is well established in the main text that isolated segments can generate propagating contractions of smooth muscle. However, the precise mechanisms involving the enteric neural circuitry remain unclear.

Agreed

Figures:

1. Figure 1 and 2 Fi and Fii: could benefit from a timeline added to the figure rather than referring to figure E and a box around the area which is magnified to make the figure clearer.

Completed.

2. Figure 5B, Title of plot. “during to Propulsion” change to “during Propulsion”.

Completed.

3. Fig 6: please check if 200uV scale bar on IJP trace is correct as looks to be same size as fig 4b.

It is correct. Please not 4b refers to EJPs and 6 is IJPs. There is considerable variability in extracellular recordings of this nature.

4. Figure 7, Legend. There are no red lines.

Corrected.

5. Figure 7. No data plot for Atropine at 50mm.

No. That is correct. We didn't obtain data at this distance for atropine, for technical reasons.

6. Figure 8: Suggest label y axis as percentage of events or CMCs

Good point. Corrected. Thanks.

7. Figure 9D: suggest adding a line and star to indicate significant result of distal colon. Also, please indicate that black bars refer to fluid distension in the figure.

Good point. Corrected now. We mention that the black bars refer to fluid distension.

8. Figure 10: reword "The major discovery is that long way ..." also rephrase "is not because the neural activity hasn't reached..." (double negative and informal language). The final 2 sentences of this fig legend also need to be clarified.

Good point. Corrected now.

9. Supplementary Figure 2, Legend. "F, shows the period represented by the dotted bar in C on expanded time..." change to "F, shows the period represented by the dotted bar in E on expanded time..."

Corrected. This is now Supplementary Figure 3. Thanks for such careful reviewing.

Reviewer #3 (Remarks to the Author):

Thank you very much to Reviewer 3. Excellent comments were raised. We have included all suggestions and made all modifications.

In this paper, Spencer and colleagues combined video imaging of colonic motility with smooth muscle electrophysiology recordings at two sites of varying distances to demonstrate synchronized rhythmic neurogenic motor activity along the length of the colon during fluid propulsion and maintained distension with a rod inserted into the lumen. This study follows on from their previous work demonstrating that large populations of enteric neurons fire in coordinated and repetitive bursts which generate rhythmic electrical activity in the smooth muscle.

1. In the introduction, it is mentioned that CMMCs originally described in mouse colon is present in other species. It would be helpful to actually refer to original work in other species as the ones cited (38, 39) refer to studies conducted in mouse.

Sure. We have cited now humans and guinea-pigs.

2. In the results, sometimes the number of trials performed, or number of contractions assessed is unclear. For instance, for the p-values presented in tables 1 and 2 of the supplementary results, how many trials or how many contractions were examined for each distance of electrode separation and per condition?

This information has now been added to each table. Good point. See newly revised Supplementary Tables 2 & 3.

3. It may be informative to show all the individual data points in the box plots in the figures.

We have tried this, but the resulting plots were not clearer.

4. It is described in the results that all spatial and temporal coordination of EJPs and IJPs was abolished by hexamethonium and tetrodotoxin. It would be interesting to also see this data presented.

Absolutely. We have included this in new Figure 9 and described the pattern of electrical activity induced in TTX in the results. Also, as requested, we updated the Figures 5B, 6A & B and 8A and newly revised Table 2 and 3.

5. Figure 6B shows electrical recordings in the presence of nicardipine + atropine. Can the authors please clarify if the atropine data presented throughout the manuscript actually refer to recordings performed in the presence of both blockers or atropine alone?

Good question. This was unclear. The recordings in atropine were always made with nicardipine added prior. The Tables (Tables 2 & 3) have been revised to make this clear.

6. On page 11, the summary of the L-NOARG analysis should refer to Table 3, not Table 2.

Thank you. This new table of analysis is presented in the new Table, Table 4 – which deals exclusively with the p Values for L-NOARG data in proximal and distal colon.

7. The authors reason that the delay in muscle contraction downstream of the advancing contraction may be explained by a concurrent activation of descending inhibitory nerve pathways over long ranges. To test this, L-NOARG was used to block the synthesis of the major inhibitory neurotransmitter nitric oxide and showed that this significantly increased the number of action potentials in the distal colon while the proximal colon was unaffected. Could the authors also show or comment on the effect of L-NOARG on the propagation of contractions and the temporal delay? Does the colon contract simultaneously in the absence of descending inhibition as predicted?

Yes, the temporal delay in propagation of the mechanical correlate of CMCs (that underlie propulsion) is eliminated by L-NOARG. This was actually shown in Fida R et al. (1997) *Neurogastroenterol & Motil*. But, at that time, we had no idea of the underlying electrical activities in the muscle occurring over large spatial fields, nor did we know the ENS fires in coordinated

discharges at ~2Hz during propulsion.

8. The final paragraph of the results describes a cohort of experiments where synchronized EJPs discharge along the whole colon, without uniform distension and without propulsion. In how many instances was this observed?

This was observed in 3 mice. This is now included in the Results, which corresponds to Movie 4.

9. It is proposed in the discussion that "...immediately prior to, and during the aboral propulsion of content, there is simultaneous activation of excitatory and inhibitory motor neurons, by a shared chain of interneurons". However, it appears that during CMCs IJPs discharged at a lower frequency (~1Hz) compared to that of EJPs (~2 Hz). If the two populations of motor neurons are indeed driven by common inputs from shared interneurons as depicted in Figure 10, would it not be expected that both IJPs and EJPs ought to display a similar discharge frequency? If this interpretation is correct, then can the authors please provide an explanation for this disparity?

Good question. There is some variability of coordinated IJPs in atropine. In 2005, we found IJPs spatially and temporally coordinated at ~2Hz immediately prior to CMCs in mouse colon, see Fig.6C in Spencer N et al. 2005; J. Physiol. So, IJPs do show a similar frequency to EJPs. In atropine, CMCs do slow down in frequency in general, see Fida et al. 1997. We found here that atropine did slow down the discharge pattern of IJPs to around 1Hz. We suspect this is because of blockage of slow muscarinic synaptic transmission in the ENS. The reason why only some (not all) IJPs slow down to 1Hz, is unclear.

10. In the discussion, the following sentence needs rephrasing: "It is known that the proximal colon displays neurogenic motor patterns that are unique to the proximal colon, including unique enteric circuitry."

Corrected.

11. Might the title of Figure 10 ("Intrinsic neural circuit identified that underlies propagating neurogenic contractions along the colon") be reconsidered since the model proposed does not actually account for how this circuit may drive propagating contractions into a downstream region of active inhibition (as is also stated in the discussion on page 15)? Further, the "repetitive activation of inhibitory and excitatory motor neurons..." described is not uniquely observed during propulsive contractions but is also observed during non-propulsive contractions. How the coordinated and repetitive activation of this enteric circuit can give rise to both propagating and non-propagating motor patterns is unclear. Perhaps the authors can elaborate on this in the discussion.

Good question. In answer to your question about propagating and non-propagating propulsive contractions, the supplementary figures and movies show that during a non-propulsive contraction (where half the colon remains contracted and the other half remained distended), the repetitive activation of EJPs still occurs. We suspect that there is only one major hard-wired circuit underlying CMCs in mouse colon which is activated by maintained distension, or by moving propulsive contractions. This is based on the finding that irrespective of which event occurs, the 2Hz pattern still occurs. But, from the work of Li et al. 2019 paper (Elife), there are different pathways and

patterns of motor activity in the proximal colon. From our experience, the major CMC circuit can be activated and manifest in many ways, such as how far CMC circuits can propagate along the colon and in what direction that choose to propagate. We have changed the title of new Figure 12 to state: **“Intrinsic neural circuit identified that underlies propagating and non-propagating neurogenic contractions along the colon”**

12. Figure 8 shows that about 11% of events examined with electrodes attached were orally directed propagating contractions. Were these events also analyzed and if so, was synchronous electrical activity along the length of the colon also observed during these contractions?

We didn't specifically analyse the degree of correlation of EJPs in orally propagating events. Our subjective impression is they also utilize ~2Hz EJPs synchronization also along the length of colon. In these examples, ascending excitatory pathways dominate and descending inhibition does not mask the EJPs from reaching action potential threshold. This would require extensive experimentation and will form the basis of future studies.

13. In the methodology for spike detection section, the following sentence needs revision: “...in which a template for the template is formed...”

Corrected

14. It would be helpful to show labels for the diameter scale in the DMaps in Figures 1B and 2B.

Good point. Included now.

15. The legend for Figure 7 refers to a red line that is notably absent.

Thanks a lot for noticing this. We have deleted this comment.

16. For reference it would also be good to have also some examples of the coherence, correlation and autocorrelation plots from the intestines that were treated with TTX and HEX. Furthermore it was mentioned that in HEX and TTX, synchrony was absent, but what happened to the signals? They were mostly gone I suppose ?

Good point. We have added a new figure 9 and analysed new data (which is included). This shows raw data in TTX and coherence (or lack thereof) of smooth muscle action potentials. In TTX, CMCs ceased and action potentials that occur are myogenic - they propagate shorter distances. We have included in the results the cumulative data now from the 4 mice that were exposed to TTX. We describe myogenic bursts of smooth muscle action potentials in the Results section now, that occur every ~31s with a mean duration of 17s.

17. In the supplementary figures it is clear that there is a lot of variation between preparations. This is of course normal and to be expected. In this respect it would be informative to see some data on how consistent the measurements and advanced analysis are. Do repetitive stimuli generate very similar data and correlation plots? Is the variability mainly carried by repeats, animals, and what happens over time, was there any influence of sex as both female and male mice were used. In this

respect it should also be made very clear what N= x actually means. Different measurements, different animals....

Good question. We have made it clear now what “N” refers to in the revised Tables. We have gone through each recording from each animal and revised the three tables (Tables 1-4) to show precisely how many recordings were made from how many (N) mice. We don’t make recordings specifically from only males or females, which would bias the data; we made recordings from both sexes. Hence, we have not segregated specifically analysis of males from females. Previously studies have not revealed pronounced changes between the two sexes. Regarding the question about whether repetitive stimuli generate very similar data and the correlation plots: we recently showed there is a high degree of reproducibility in terms of ability to elicit propulsion using pulses of electrical nerve stimuli – See Barth et al. 2021); Am J Physiol: [10.1152/ajpgi.00463.2020](https://doi.org/10.1152/ajpgi.00463.2020). With fluid distension, this is difficult to quantify with variable volumes of fluid expelled and required to reach threshold for propulsion.

REVIEWERS' COMMENTS:

Reviewer #1 (Remarks to the Author):

The authors have adequately addressed my comments. It is good for publication in my opinion.

Reviewer #2 (Remarks to the Author):

I recommend accepting the revised manuscript since the authors have responded to all the points raised.

Reviewer #3 (Remarks to the Author):

Thank you for the changes and clarifications.

Please change p-vals (p-values) and Kreb's, the latter being either Krebs or Krebs', but not Mr. or Mrs. Kreb's solution.